# Gut-Testis Axis: Microbiota Prime Metabolome To Increase Sperm Quality in Young Type 2 Diabetes

Xiaowei Yan,[a,b] Yanni Feng,[c] Yanan Hao,[a,b,d] Ruqing Zhong,[a] Yue Jiang,[a] Xiangfang Tang,[a] Dongxin Lu,[a,b] Hanhan Fang,[a,b] Manjree Agarwal,[d,e] Liang Chen,[a] ⓘ Yong Zhao,[a] Hongfu Zhang[a]

aState Key Laboratory of Animal Nutrition, Institute of Animal Sciences, Chinese Academy of Agricultural Sciences, Beijing, People's Republic of China
bCollege of Life Sciences, Qingdao Agricultural University, Qingdao, People's Republic of China
cCollege of Veterinary Medicine, Qingdao Agricultural University, Qingdao, People's Republic of China
dCollege of Science, Health, Engineering and Education, Murdoch University, Perth, Australia
eScientific Service Division, ChemCentre, Government of Western Australia, Bentley, Australia

Xiaowei Yan, Yanni Feng, Yanan Hao, Ruqing Zhong, and Yue Jiang contributed equally to this article. Author order was determined by drawing straws.

**ABSTRACT** Young type 2 diabetes (T2D) affects 15% of the population, with a noted increase in cases, and T2D-related male infertility has become a serious issue in recent years. The current study aimed to explore the improvements of alginate oligosaccharide (AOS)-modified gut microbiota on semen quality in T2D. The T2D was established in young mice of 5 weeks of age with a blood glucose level of 21.2 ± 2.2 mmol/L, while blood glucose was 8.7 ± 1.1 mM in control animals. We discovered that fecal microbiota transplantation (FMT) of AOS-improved microbiota (A10-FMT) significantly decreased blood glucose, while FMT of gut microbiota from control animals (Con-FMT) did not. Sperm concentration and motility were decreased in T2D to 10% to 20% of those in the control group, while A10-FMT brought about a recovery of around 5- to 10-fold. A10-FMT significantly increased small intestinal *Allobaculum*, while it elevated small intestinal and cecal *Lactobacillus* in some extent, blood butyric acid and derivatives and eicosapentaenoic acid (EPA), and testicular docosahexaenoic acid (DHA), EPA, and testosterone and its derivatives. Furthermore, A10-FMT improved liver functions and systemic antioxidant environments. Most importantly, A10-FMT promoted spermatogenesis through the improvement in the expression of proteins important for spermatogenesis to increase sperm concentration and motility. The underlying mechanisms may be that A10-FMT increased gut-beneficial microbes *Lactobacillus* and *Allobaculum* to elevate blood and/or testicular butyric acid, DHA, EPA, and testosterone to promote spermatogenesis and thus to ameliorate sperm concentration and motility. AOS-improved gut microbes could emerge as attractive candidates to treat T2D-diminished semen quality.

**IMPORTANCE** A10-FMT benefits gut microbiota, liver function, and systemic environment via improvement in blood metabolome, consequently to favor the testicular microenvironment to improve spermatogenesis process and to boost T2D-diminished semen quality. We established that AOS-improved gut microbiota may be used to boost T2D-decreased semen quality and metabolic disease-related male subfertility.

**KEYWORDS** type 2 diabetes in youth, sperm concentration, sperm motility, spermatogenesis, gut microbiota, DHA, EPA

Address correspondence to Yong Zhao, yzhao818@hotmail.com.

The authors declare no conflict of interest.

Type 2 diabetes (T2D) has long been considered a disease of adulthood; however, there has been a steep rise of this disease in children and adolescents worldwide (1–5), parallel to the increasing rates of obesity. It is reported the incidence of T2D in children <15 years of age in New Zealand has increased progressively at 5% per year over the past 2 decades (2). The reasons for this increase include genetic factors,

environmental factors, and a lack of physical activity (1–5). T2D patients are at potentially higher risk for most of the outcomes compared to type 1 diabetes (T1D) patients. For example, 72% of T2D patients display different types of early diabetes-related complications; however, just 32% of T1D patients have similar evidence (3). The most impressive feature of T2D in youth is that it is more commonly diagnosed at lower age and a lower body mass index in boys than in girls (4). Moreover, the majority of youths with T2D present at a mean age of 13.5 years (1–5).

Accumulating data from human and animal studies indicate that gut bacteria play fundamental roles in T2D, as there is profound dysbiosis in T2D (6–9). Gut microbiota and hosts have developed a coherent symbiotic relationship where the gut microbiota contribute to many physiological functions, including energy metabolism, metabolic signaling, immune system formation, and gut barrier integrity (9). Gut microbiota benefits humans through the production of short-chain fatty acids (SCFAs) from the fermentation of carbohydrates; moreover, a deficiency in SCFAs is correlated with T2D (8). *Bifidobacterium*, *Bacteroides*, *Faecalibacterium*, *Akkermansia*, and *Roseburia* are reported to be negatively associated with T2D; however, *Ruminococcus*, *Fusobacterium*, and *Blautia* are positively associated. *Lactobacillus* is frequently detected and reported in T2D and has produced inconsistent results among investigations (7, 9). In humans and experimental animal models, *Bacteroides* has been shown to be a beneficial microbe for glucose metabolism (6). Moreover, microbes have been considered a treatment for T2D (7), especially as dietary intervention can modulate the gut microbiota to improve glucose status in T2D (7).

Male infertility is already a common health problem worldwide, and up to one in six couples have infertility issues (10, 11). Diabetes poses an adverse impact on male fertility directly and indirectly at multiple levels, such as impairment to spermatogenesis itself, penile erection, and ejaculation (12–15). Thus, male infertility issues have become more common with the increasing rates of T2D. The negative impacts of diabetes on erectile and ejaculation function, as well as a reduction in semen volume, sperm count, sperm motility, and abnormal sperm morphology, have been reported widely (13). Since T2D-related male infertility is such a serious issue, much effort has been directed to overcome it. The hydroalcoholic extract of *Rhus coriaria* seeds, resveratrol, metformin, and chitosan-stabilized selenium nanoparticle combination, adiponectin, nesfatin-1, and testosterone have been used in the treatment of T2D-impaired semen quality and related male infertility (16–22). Recently, our group and others have found that gut dysbiosis disrupts spermatogenesis to decrease semen quality and/or male fertility (23–26). Moreover, gut microbiota transplantation (FMT) has been shown to be an effective approach for improving semen quality (23, 24). However, it is not understood whether gut microbiota could effectively ameliorate semen quality in T2D subjects. The current study aimed to explore the beneficial improvement of alginate oligosaccharide (AOS)-modified gut microbiota on semen quality in T2D since it has been found to be effective in ameliorating semen quality in busulfan-treated subjects (23, 24).

## RESULTS

**A10-FMT reduced blood glucose, increased sperm concentration and sperm motility, and improved gut microbiota in T2D.** Three-week-old male mice were fed with a high-fat diet (HFD) for 2 weeks, after which one dose of streptozotocin (STZ; 85 mg/kg body weight) was injected intraperitoneally (i.p.). After 3 days, blood glucose (21.2 ± 2.2 mmol/L [mM]) was significantly higher in the HFD plus STZ (HS) group than in the control (Con) group (8.7 ± 1.1 mM), which indicated that a T2D model has been successfully established (blood glucose > 11.1 mM) (27, 28). Then, mice in the HS group were divided into three groups, (i) the HS group (STZ+HFD), (ii) the A10-FMT group (HFD+STZ with fecal microbiota transplantation [FMT] from AOS-improved gut microbiota), and (iii) Con-FMT group (HFD+STZ plus FMT from control animal gut microbiota) (see Fig. 1a for study scheme and see Materials and Methods for details). After another 5 weeks, the body weight and blood insulin levels were lower in the HS, A10-FMT, and Con-FMT groups than that in the Con group (see Fig. S1a and b in the

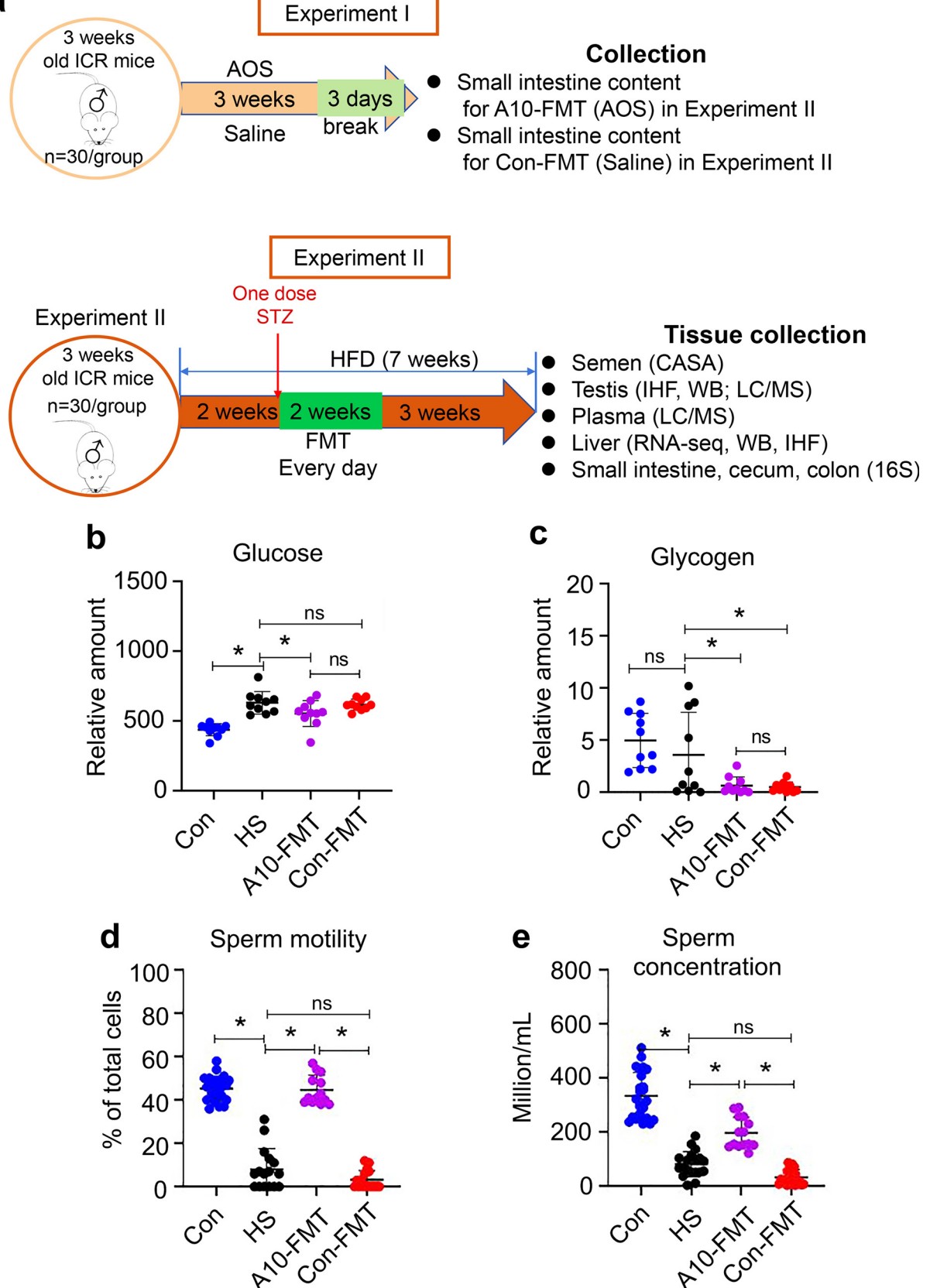

**FIG 1** A10-FMT decreased blood glucose and improved semen quality. (a) Study scheme. (b) Blood glucose levels determined by LC/MS. The $y$ axis represents the relative concentration. The $x$ axis represents the treatment. (c) Blood glycogen levels determined by LC/MS.

supplemental material). Blood glucose was significantly higher in the HS group than that in the Con group; however, it was significantly reduced by A10-FMT, but not by Con-FMT (Fig. 1b), which indicated that A10-FMT treatment improved T2D status. At the same time, blood glycogen was decreased by both A10-FMT and Con-FMT (Fig. 1c). Sperm concentration and motility were significantly diminished in T2D subjects (HS group), while A10-FMT (not Con-FMT) significantly increased sperm motility and concentration (Fig. 1d and e).

Gut microbiota has been reported to be disturbed in T2D patients and animal models (6, 9). Small intestine, cecum, and colon content microbiota were determined in the current investigation (Fig. 2; Fig. S1c to n; Fig. S2). In the small intestine, *Allobaculum* was increased in the A10-FMT group compared to Con and HS groups (Fig. 2a to d), while *Desulfovibrio* was decreased in the HS, A10-FMT, and Con-FMT groups compared to Con (Fig. 2a to d). In the cecum, *Bacteroides* were increased in the A10-FMT and Con-FMT groups compared to Con or HS (Fig. S2a to d), while both *Coprococcus* and *Flexispira* were decreased in the HS, A10-FMT, and Con-FMT groups compared to Con (Fig. S2a to d); however, *Helicobacter* was increased in the HS group compared to Con, while it was decreased in the A10-FMT and Con-FMT groups (Fig. S2a to d). On the other hand, *Lactobacillus* was decreased in the HS, while it was increased in the A10-FMT and Con-FMT groups (Fig. S2a to d), although not significantly. In the colon, *Bacteroides* and *Helicobacter* were altered in the same trend as in the cecum (Fig. S2e to h).

The function of changed gut microbe genes was enriched by Kyoto Encyclopedia of Genes and Genomes (KEGG) analysis, and eight major signaling pathways were disturbed by HS, while they were reversed in the A10-FMT and/or Con-FMT groups in the small intestine, cecum, and/or colon (Fig. 2e). Interestingly, the energy metabolism pathway was increased in the colon, while it was decreased in the small intestine and cecum in the A10-FMT group specifically, and the lipid metabolism pathway was decreased in the A10-FMT group, while it was increased in the Con-FMT group in the colon (Fig. 2e). However, the glycan biosynthesis and metabolism and metabolism of cofactor and vitamins pathways were decreased in the HS group, while they were increased in both the A10-FMT and Con-FMT groups in the cecum and colon (Fig. 2e). The membrane transport pathway was increased by HS, while it was decreased in both the A10-FMT and Con-FMT groups in the colon (Fig. 2e), and the cell motility pathway was increased by HS, while it was decreased in both the A10-FMT and Con-FMT groups in the cecum (Fig. 2f). Moreover, the carbohydrate metabolism pathway was decreased by HS, while it was increased in the A10-FMT and Con-FMT groups in the small intestine. On the other hand, the carbohydrate metabolism pathway was increased by HS, while it was decreased in the A10-FMT and Con-FMT groups in the colon (Fig. 2e). In total, the data indicated that A10-FMT and Con-FMT may differentially modulate gut microbiota and microbial function to regulate blood metabolites and other functions such as spermatogenesis in type 2 diabetes.

**A10-FMT ameliorated blood metabolome and liver functions in T2D.** T2D is a metabolic disease. Gut microbiota regulates the blood metabolome, which was confirmed in the current study by liquid chromatography-mass spectrometry (LC-MS) (Fig. S3; Data Set S1). Compared to the Con group, blood triglyceride (TG) was increased in the HS group, which was decreased in the A10-FMT group; however, the data were not significant. Cholesterol was higher in the HS group, while it was decreased in the A10-FMT group (Fig. 3a and b). At the same time, blood lipid molecules were increased in the HS group; however, they were reduced by A10-FMT and/or Con-FMT (Fig. 3c and d), which is consistent with gut microbiota data. Bile acids play vital roles in lipid and other metabolism in the intestine and liver (25). The most common bile acids were increased in the HS and Con-FMT groups, while they were decreased in the A10-FMT group (Fig. 3e to g). It is very interesting to note that butyric acids and derivatives were decreased in the HS group, while they were increased in the A10-FMT

**FIG 1** Legend (Continued)
The *y* axis represents the relative amount. The *x* axis represents the treatment. (d) Sperm motility. The *y* axis represents the percentage of cells. The *x* axis represents the treatment. (e) Sperm concentration. The *y* axis represents sperm concentration. The *x* axis represents the treatment. $n = 30$/group, *, $P < 0.05$; ns, not significant.

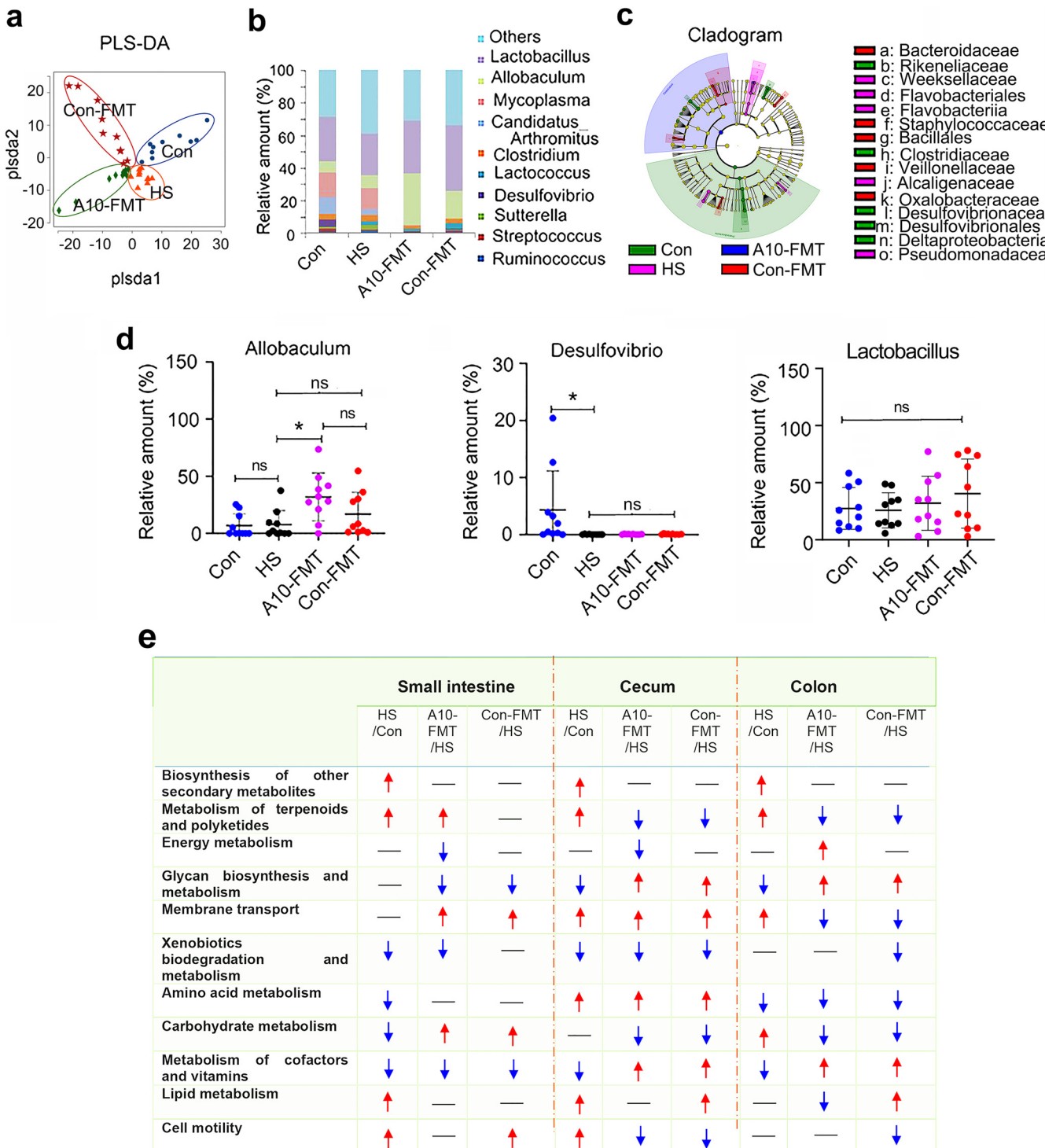

**FIG 2** A10-FMT improved small intestinal microbiota in type 2 diabetes. (a) PLS-DA (OTU) of small intestine microbiota in HS, A10-FMT, and Con-FMT groups. (b) Small intestine microbiota levels at the genus level in HS, A10-FMT, and Con-FMT groups. The y axis represents the relative amount (%). The x axis represents the treatments. Different colors represent different microbiota. (c) Cladogram of the linear discriminate analysis effect size (LEfSe) determining the difference in abundance of small intestine microbiota. (d) *Allobaculum, Desulfovibrio*, and *Lactobacillus* in the small intestine. The y axis represents the relative amount at the genus level. The x axis represents the treatment. *, P < 0.05. (e) Summary of signaling pathways of changed microbiota genes by Kyoto Encyclopedia of Genes and Genomes (KEGG) enrichment analysis. Red arrow indicates increased microbiota genes in each comparison. Blue arrow indicates decreased microbiota genes in each comparison. n = 10/group, ns, not significant.

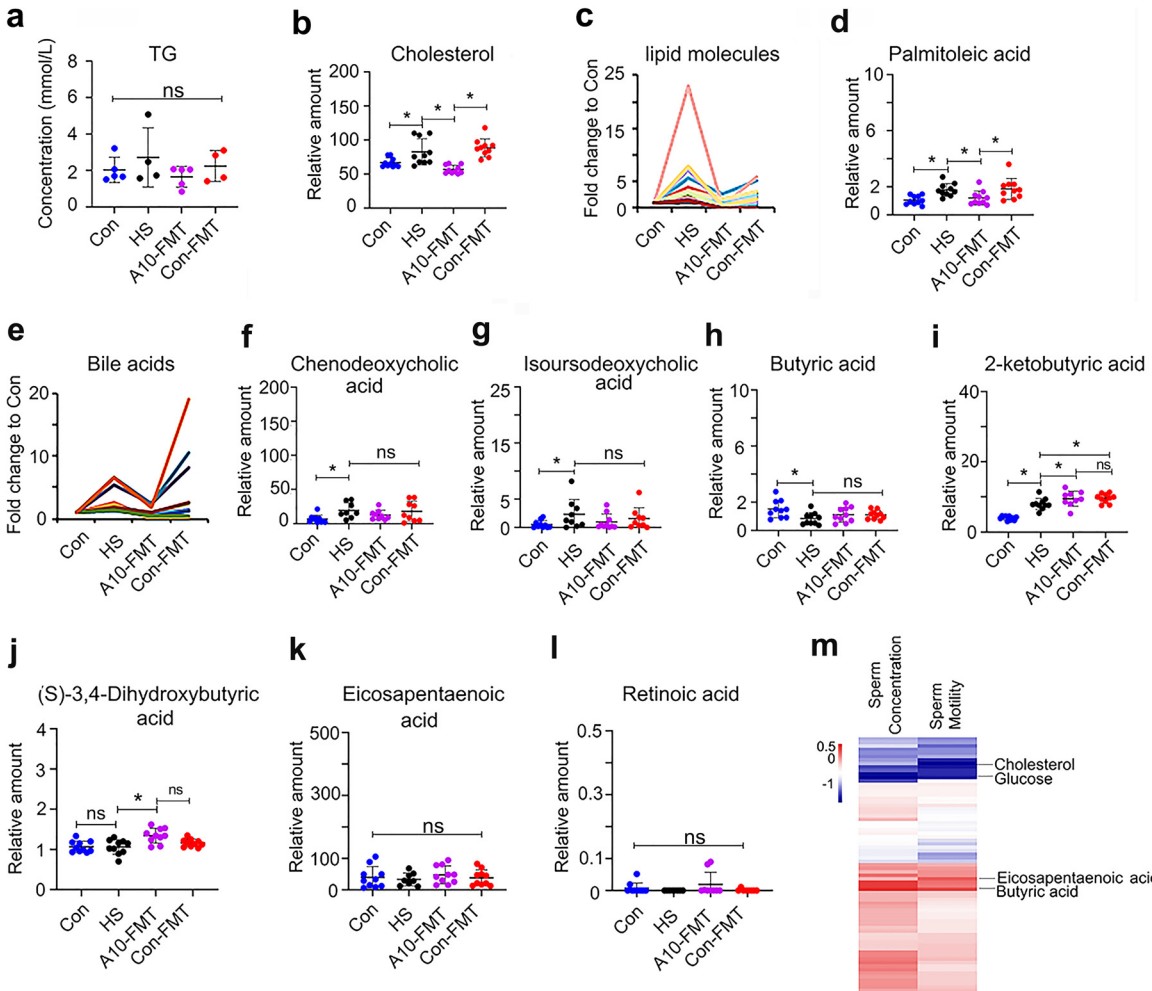

FIG 3 A10-FMT improved blood metabolism (blood TG was determined by the specific kit, while others were quantified by LC/MS). (a) Blood TG levels in different treatments. The *y* axis represents the concentration (mmol/L). The *x* axis represents the treatment. (b) Blood cholesterol levels in different treatments. (c) Blood lipid molecules levels in different treatments. The *y* axis represents the fold change to Con. The *x* axis represents the treatment. (d) Blood palmitoleic acid levels in different treatments. (e) Blood bile acid levels in different treatments. The *y* axis represents the fold change to Con. The *x* axis represents the treatment. (f) Blood chenodeoxycholic acid levels in different treatments. (g) Blood isoursodeoxycholic acid levels in different treatments. (h) Blood butyric acid levels in different treatments. (i) Blood ketobutyric acid levels in different treatments. (j) Blood (S)-3,4-dihydroxybutyric acid levels in different treatments. (k) Blood eicosapentaenoic acid (EPA) levels in different treatments. (l) Blood retinoic acid levels in different treatments. (m) Correlation of blood metabolites and sperm motility and sperm concentration. (The data from all groups for the blood metabolites and sperm motility and concentration were used to perform the correlation analysis). For panels b, d, and f to l, the *y* axis represents the relative amount, and the *x* axis represents the treatment, *, $P < 0.05$ for all ($n = 10$/group); ns, not significant.

group (Fig. 3h to j). The blood *n*-3 polyunsaturated fatty acids (PUFA) and eicosapentaenoic acid (EPA) were lower in the HS group, while they increased in the A10-FMT group (Fig. 3k). Similarly, blood retinoic acid was in a decreasing trend in the HS group, while it was in an increasing trend in the A10-FMT group; however, the data were not significant (Fig. 3l). It should be noted that in some cases, the data were not significant between the Con and HS groups (Fig. 3j to l), which suggests that the changes observed may not be representative for T2D. The correlation analysis of blood metabolites and sperm concentration/motility showed that blood glucose and cholesterol were negatively correlated with sperm concentration and sperm motility, while EPA and butyric acid were positively correlated with sperm concentration and motility (Fig. 3m), which suggests that the blood metabolome and semen quality parameters are correlated.

The liver is the major metabolic organ that is responsible for most of the metabolites in the blood. Liver function may be disturbed by HFD and STZ treatment (HS), while A10-FMT reverses the damage, as shown by the histopathology of the liver samples in different

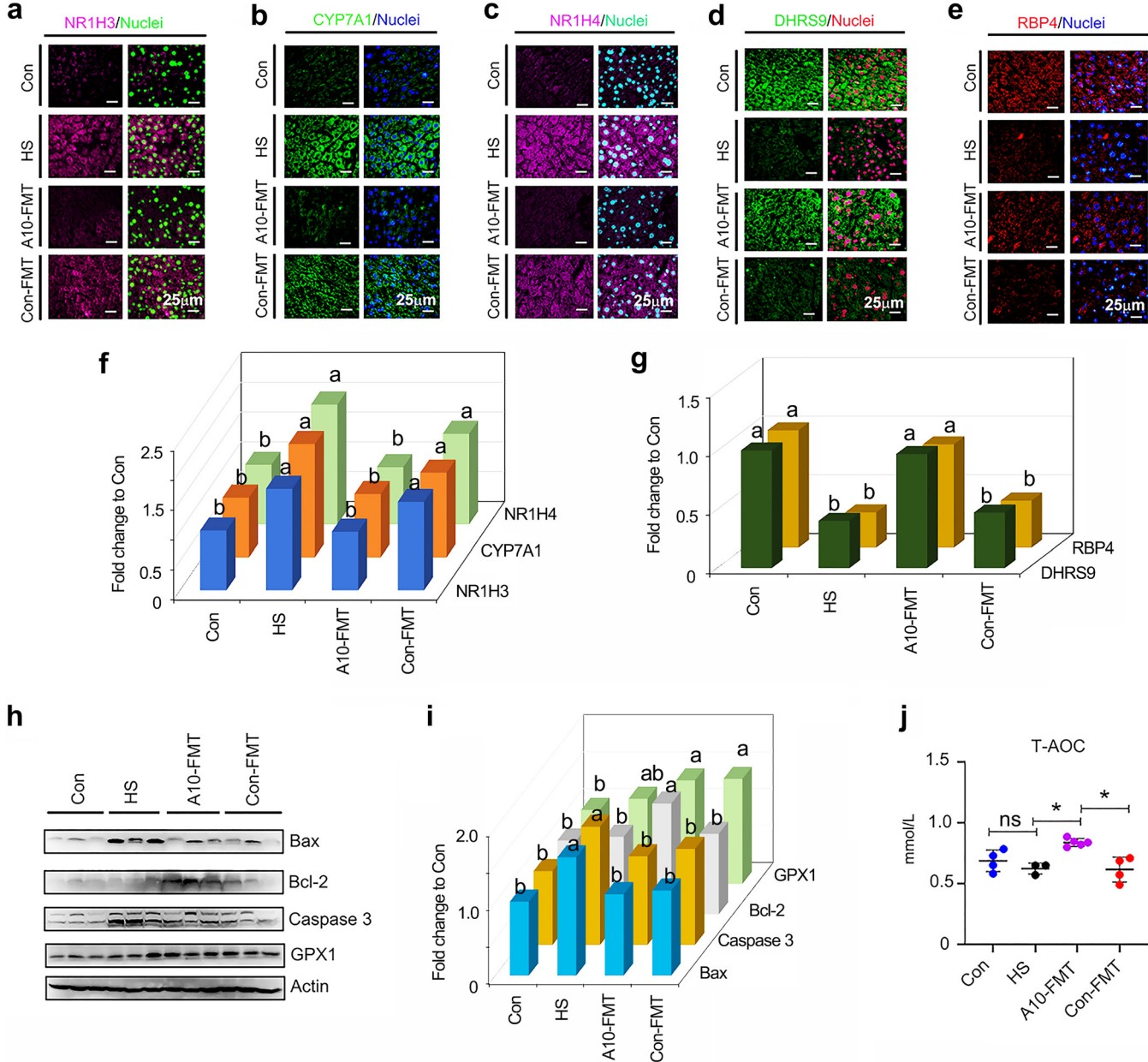

**FIG 4** A10-FMT improved liver lipid metabolism and antioxidant capability. (a) Immunofluorescence (IHF) staining of NR1H3 in liver tissue. (b) IHF staining of CYP7A1 in liver tissue. (c) IHF staining of NR1H4 in liver tissue. (d) IHF staining of DHRS9 in liver tissue. (e) IHF staining of RBP4 in liver tissue. (f) Quantitative data for IHF staining of NR1H3, CYP7A1, and NR1H4 in liver tissue. (g) Quantitative data for IHF staining of DHRS9 and RBP4 in liver tissue. (h) Western blotting of Bax, Bcl-2, caspase 3, and GPX1 in the liver. (i) Quantitative data for Western blotting of Bax, Bcl-2, caspase 3, and GPX1 in the liver. (j) Blood total antioxidant capability (T-AOC) levels. The y axis represents the concentration (mmol/L). The x axis represents the treatments. *, $P < 0.05$ ($n \geq 3$/group). For panels f, g, and i, the y axis is the fold change to the Con group. The x axis represents the treatments. a and b, significance between different groups at $P$ values of $<0.05$ for each protein ($n \geq 3$/group), ns means not significant.

groups (Fig. S4a). The transcriptome sequencing (RNA-seq) analysis of liver samples showed that HS disrupted gene expression in the liver, while it was recovered by A10-FMT and/or Con-FMT (Fig. S4b to d). Functional enrichment analysis found that liver lipid metabolism function was upset by HS (Fig. S4c). Expression of the lipid metabolism-related genes (such as *NR1H3*, *Acaa1a*, *Acaa2*, *Acox1*, *Acox2*, etc.) was decreased by HS, while it increased was by A10-FMT and/or Con-FMT (Fig. S4c). Nuclear receptor subfamily 1 group H member 3 (*NR1H3*), a key regulator of lipid homeostasis, has been found to be involved in lipid deposition in pig (29). The protein level of NR1H3 was increased in the HS group, while it was decreased in the A10-FMT group but not in the Con-FMT group (Fig. 4a and f).

Blood bile acids were significantly higher in the HS and Con-FMT groups, which was consistent with the protein levels of the bile acid-produced enzyme CYP7A1 (Fig. 4b and f) and bile acid receptor NR1H4 (Fig. 4c and f), which were higher in the HS and Con-FMT groups, while they were reduced in the A10-FMT group. Retinoic acids play vital roles in lipid metabolism in the liver (29). The protein levels of retinoic synthesis protein DHRS9 and binding protein RBP4 were decreased by HS, while they were increased by A10-FMT (Fig. 4d, e, and g). All data together indicated that liver lipid metabolism function was disrupted by the HS group and recovered by A10-MFT, which matched the blood metabolism data. Moreover, HS caused apoptosis in liver cells, which further suggested HS damaged liver function, while it was recovered by A10-FMT (Fig. 4h and i). At the same time, A10-FMT increased systemic antioxidant capability through the total antioxidant capability (T-AOC) of the blood and GPX1 in liver (Fig. 4h to j); however, the data were not significant between Con and HS (Fig. 4h to j), which suggests that the changes observed may not be representative for T2D.

**A10-FMT improved the testicular metabolome and the testicular microenvironment.** Since A10-FMT improved liver function and the blood metabolome, next, we set out to explore the effects of A10-FMT on the testicular metabolome (Fig. 5; Fig. S5; Data Set S2). *n*-3 PUFAs such as docosahexaenoic acid (DHA) and EPA play vital roles in spermatogenesis (30, 31). It is interesting to note that A10-FMT, but not Con-FMT, increased DHA and EPA and their derivatives in the testes (Fig. 5a to f). Moreover, many other unsaturated acids were also increased by A10-FMT (Fig. 5g to i) in the testes. However, there is no significant difference between the Con and HS groups for some of the compounds (Fig. 5a, c, f, i, k, and n to p), which suggests that the changes observed may not be representative for T2D. Another group of altered metabolites in the testes was retinoids. There is an increasing trend for the retinoids by A10-FMT compared to HS (Fig. 5j to p). HS significantly decreased retinyl ester and retinal, while A10-FMT significantly increased them (Fig. 5i and m). It is known that retinoids are crucial for spermatogenesis (10, 11). Male steroid hormones, especially testosterone, play vital roles in spermatogenesis. HS treatment significantly decreased testosterone and epi-testosterone, while A10-FMT led to a recovery (Fig. 6a and b). HS treatment significantly decreased androsterone, while it was in an increasing trend in A10-FMT (Fig. 6c), which, however, may not be relevant to T2D, given that the data were not significant between the Con and HS groups. The levels of hormone production proteins in the testes, which were decreased by HS, were increased by A10-FMT (Fig. 6d and e). At the same time, the changed testicular metabolites were well correlated with each other in different comparisons, including HS/Con, A10-FMT/HS, and Con-FMT/HS, respectively (Fig. S5g to i).

Furthermore, correlation analysis of testicular metabolites and sperm concentration and motility found that there was a good relationship between testicular metabolites and semen quality parameters (sperm concentration and motility [Fig. 6f]). DHA, EPA, and testosterone were positively correlated with sperm concentration and/or sperm motility (Fig. 6f). At the same time, there was a good relationship between blood metabolites and testicular metabolites (Fig. 6g), where some of the metabolites were positively correlated, while others were negatively correlated.

**A10-FMT ameliorated spermatogenesis to increase sperm concentration and motility.** The metabolome data in blood and testicular samples suggested that HS upset the systemic environment and testicular microenvironment to disrupt spermatogenesis and, in turn, reduce sperm concentration and motility; meanwhile, A10-FMT improved the systemic and testicular environment to recover spermatogenesis, sperm concentration, and motility. A10-FMT recovered spermatogenesis by increasing important spermatogenesis proteins such as VASA (germ cell marker), SYCP3 (meiosis marker), PGK2 (sperm motility and fertility protein), ODF1 (component of filamentous structure), PIWIL1 (germ line integrity), and TP1 (spermatid-specific protein) in testicular samples (Fig. 7a to f). However, there is no significant difference between the Con and HS groups for PIWIL1 and TP1; the SYCP3 data were not significant for the A10-FMT or Con-FMT groups compared to the HS group. The Sertoli cell marker, SOX9, was not changed in all the groups (Fig. 7g). The data indicate

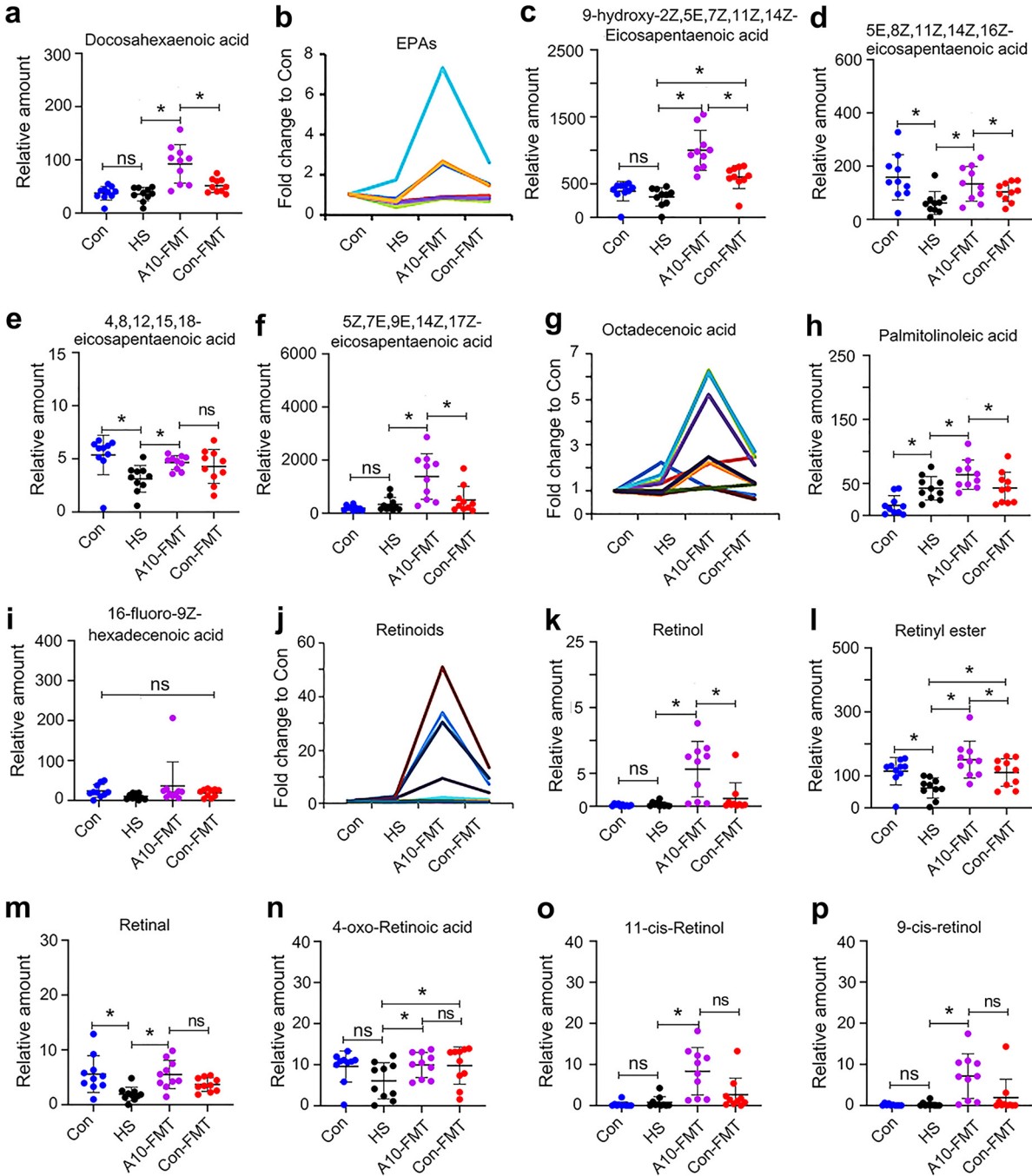

**FIG 5** A10-FMT improved testicular metabolism (determined by LC/MS). (a) Testicular docosahexaenoic acid (DHA) levels in different treatments. (b) Testicular EPAs in different treatments. The $y$ axis represents the fold change to Con. The $x$ axis represents the treatment. (c) Testicular 9-hydroxy-2Z,5E,7Z,11Z,14Z-eicosapentaenoic acid levels in different treatments. (d) Testicular 5E,8Z,11Z,14Z,16Z-eicosapentaenoic acid levels in different treatments. (e) Testicular 4,8,12,15,18-eicosapentaenoic acid levels in different treatments. (f) Testicular 5Z,7E,9E,14Z,17Z-eicosapentaenoic acid levels in different treatments. (g) Testicular octadecenoic acid levels in different treatments. The $y$ axis represents the fold changes to Con. The $x$ axis represents the treatment. (h) Testicular palmitolinoleic acid levels in different treatments. (i) Testicular acid levels in different treatments. (j) Testicular retinoids levels in different treatments. The $y$ axis represents the fold change to Con. The $x$ axis represents the treatment. (k) Testicular retinol levels in different treatments. (l) Testicular retinyl ester levels in different treatments. (m) Testicular retinal levels in different treatments. (n) Testicular 4-oxo-retinoic acid levels in different treatments. (o) Testicular 11-*cis*-retinol levels in different treatments. (p) Testicular 9-*cis*-retinol levels in different treatments. For panels a, c to e, f, h, i, and k to p, the $y$ axis represents the relative amount, and the $x$ axis represents the treatment, *, $P < 0.05$ for all ($n = 10$/group); ns, not significant.

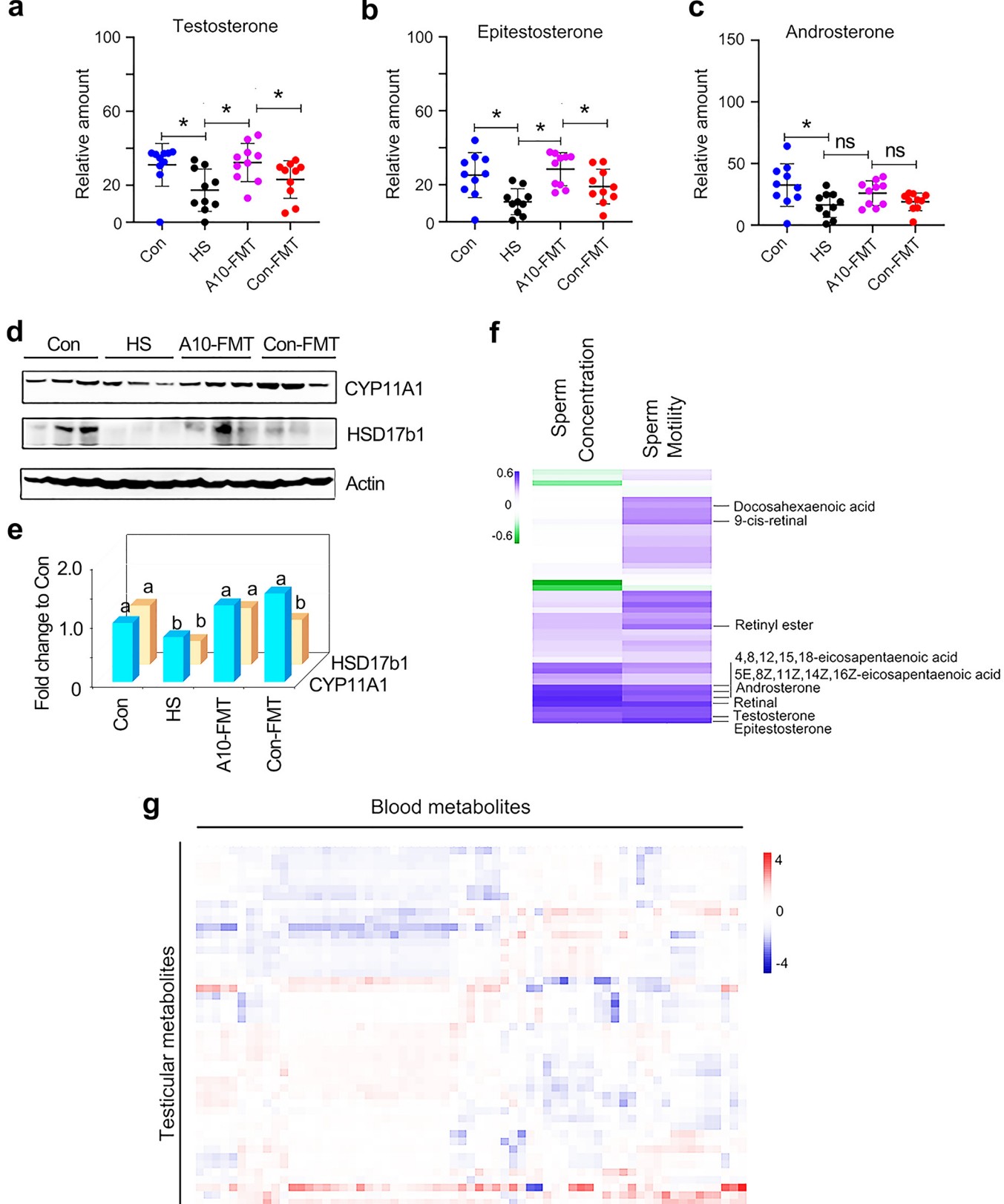

FIG 6 A10-FMT benefited testicular testosterone levels and synthesis to improve the testicular microenvironment. (a) Testicular testosterone levels in each treatment determined by LC/MS. (b) Testicular epitestosterone levels in each treatment determined by LC/MS. ns, not significant. (c) Testicular androsterone levels in each treatment determined by LC/MS. (d) Western blotting of testicular important proteins for steroid hormone production in each

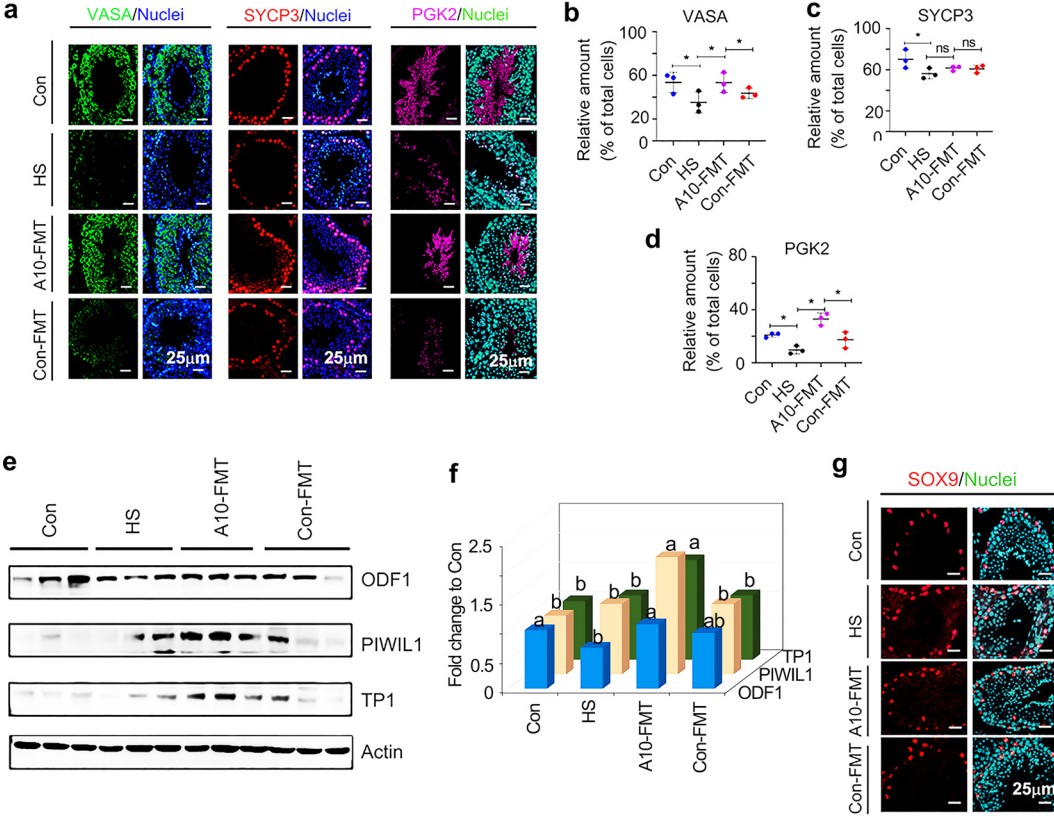

**FIG 7** A10-FMT improved spermatogenesis. (a) IHF staining of testicular germ cell marker VASA, meiosis marker SYCP3, and sperm protein PGK2 in each treatment. Scale bar, 50 $\mu$m. (b) Quantitative data for VASA IHF staining. The $y$ axis represents the percentage of total cell. The $x$ axis represents the treatment. *, $P < 0.05$. (c) Quantitative data for SYCP3 IHF staining. The $y$ axis represents the percentage of total cell. The $x$ axis represents the treatment. *, $P < 0.05$. (d) Quantitative data for PGK2 IHF staining. The $y$ axis represents the percentage of total cell. The $x$ axis represents the treatment. *, $P < 0.05$. (e) Western blotting of the proteins of important genes for spermatogenesis in each treatment. (f) Quantitative data for Western blotting of ODF1, PIWIL1, and TP1 in the testis. The $y$ axis was the fold change to Con group. The $x$ axis represents the treatments. a and b, significant between different groups at $P$ values of $<0.05$ for each protein ($n \geq 3$/group). (g) IHF staining of Sertoli cell marker SOX9. For panels b to d, the $y$ axis represents the percentage of total cell, and the $x$ axis represents the treatment, *, $P < 0.05$ for all ($n \geq 3$/group); ns, not significant.

that A10-FMT improved spermatogenesis and thus increased sperm concentration and motility.

## DISCUSSION

As more youths develop T2D, the risk of complications is high; therefore, many studies are focusing on this disease. It has been reported that T2D in youths is more frequently diagnosed at a lower age and body mass index in boys than girls (4). Furthermore, the mean age of T2D in youths is 13.5 years, which is around the initiation of puberty (1–5), a time when the male reproductive system begins rapid development. This may be the main reason for male infertility (diminished semen quality) in T2D during later reproductive age since hyperglycemia in T2D causes reproductive system dysfunction (13). In the current study, in order to mimic youth T2D patients, 3-week-old male mice were fed with HFD and injected with STZ to induce T2D, and during puberty (5 weeks of age), the mice developed T2D (32, 33). Sperm concentration and motility were decreased, while blood glucose was increased in T2D mice

**FIG 6 Legend (Continued)**

treatment. (e) Quantitative data for Western blotting of CYP11A1 and HSD17b1 in the testis. The $y$ axis was the fold change to the Con group. The $x$ axis represents the treatments. a and b, significant between different groups at $P$ values of $<0.05$ for each protein ($n \geq 3$/group). (f) Correlation of testicular metabolites and sperm concentration and sperm motility. (g) Correlation of testicular metabolites and blood metabolites. For panels a to c, the $y$ axis represents the relative amount, and the $x$ axis represents the treatment, *, $P < 0.05$ for all ($n = 10$/group); ns, not significant.

when adult (10 weeks of age), which is consistent with clinical findings (15). However, FMT from AOS-improved gut microbiota (A10-FMT) decreased blood glucose and increased sperm concentration and motility, while Con-FMT did not. The data suggest that AOS-improved gut microbiota contains beneficial microbes that can produce or help with the production of beneficial compounds to ameliorate the systemic environment and testicular microenvironment to improve spermatogenesis. In the current investigation, sperm motility and concentration were increased by A10-FMT compared to T2D (HS group); however, the pregnancy rate was not determined.

Gut microbiota plays a crucial role in T2D development (6–9). *Bacteroides* and *Bifidobacterium* are beneficial genera that are most frequently reported in studies of T2D (6). In the current study, we found that *Bacteroides* was increased by A10-FMT and Con-FMT in both the cecum and colon. Data suggest that FMT may be helpful in glucose metabolism, as it has been previously reported that *Bacteroides* beneficially supports glucose metabolism (6). Actually, blood glycogen was decreased by both A10-FMT and Con-FMT. *Helicobacter*, one of the gastric-pathogenic bacteria (34, 35), was increased in the cecum and colon in T2D mice, while it was reduced by A10-FMT and Con-FMT in the current investigation. Although *Lactobacillus* may be considered a beneficial bacterial in T2D, the data were not constant (6). *Lactobacillus* was decreased in T2D animals, while it was strongly increased by A10-FMT, although not significantly in the current investigation. Another interesting microbe, *Allobaculum*, an SCFA-producing microbe (35–39), was significantly increased by A10-FMT, while it was increased by Con-FMT (not significantly) compared to the HS group in the small intestine in the current study. At the same time, we found that the blood SCFA, butyric acid, and its derivatives were decreased in T2D and increased by A10-FMT (Con-FMT also increased them to some extent); moreover, blood butyric acid and sperm concentration and motility were positively correlated with each other. It is known that SCFA deficiency is associated with T2D, and dietary fiber can selectively promote gut bacteria to produce SCFAs to alleviate T2D (8). Our findings further confirmed that gut microbiota selectively promoted by dietary components may produce beneficial compounds to attenuate T2D and even improve sperm concentration and motility in T2D, as SCFAs act as signaling molecules in humans (8, 40, 41).

T2D is a metabolic disease (3–5). The gut microbiota is involved in metabolism to produce specific molecules in the blood to benefit systemic health (8), and metabolism plays a vital role in spermatogenesis (42). A10-FMT improved gut microbiota by increasing beneficial microbiota to then benefit blood metabolism. A10-FMT reduced blood TG, cholesterol, glucose, and glycogen, while it increased butyric acid and derivatives and EPA. Furthermore, blood glucose and cholesterol were negatively correlated with sperm concentration and motility, while EPA and butyric acid were positively correlated with sperm concentration and motility. Furthermore, A10-FMT improved T2D-impaired liver function to support systemic metabolism. The testicular microenvironment is important for spermatogenesis (42). A10-FMT improved the testicular microenvironment through the significant increase in testicular levels of DHA, EPA, and testosterone and its derivatives. *n*-3 PUFAs, especially DHA, are crucial for spermatogenesis and male reproductive functions (30, 31, 43). Moreover, testicular metabolites and blood metabolites were well correlated. All the data indicated that A10-FMT improved the systemic environment and testicular microenvironment to promote spermatogenesis to increase sperm concentration and motility. Indeed, it has previously been shown that A10-FMT improves spermatogenesis impaired by busulfan through amelioration of the important proteins in spermatogenesis (23, 24). Although many studies have tried different approaches to ameliorate semen quality in T2D subjects, it was previously unknown whether gut microbiota could improve spermatogenesis to increase semen quality in youths with T2D. For the first time, we established that AOS-improved gut microbiota ameliorated sperm concentration and motility and glucose status in youths with T2D.

In summary, this novel investigation in youthful mice with T2D demonstrated that A10-FMT ameliorated gut microbes *Lactobacillus* and *Allobaculum* to elevate blood

butyric acid and EPA and testicular DHA, EPA, and steroid hormones such as testosterone to promote spermatogenesis, thus increasing sperm concentration and sperm motility. The findings from the current study shed new light on the mechanism of action of beneficial gut microbiota in youths with T2D who have potentially high risks of male infertility. Our findings strongly suggest the need to explore microbiota therapy (FMT) to reduce the elevated rates of male infertility in youths with T2D early in their disease course. Clinical studies in youths with T2D are warranted to elucidate the roles of beneficial microbiota in the improvement of semen quality and male fertility.

## MATERIALS AND METHODS

**Study design.** All animal procedures used in this study were approved by the Animal Care and Use Committee of the Institute of Animal Sciences of Chinese Academy of Agricultural Sciences (IAS2020-106). The male mice were used in the current study due to its focus on male reproductive health. The mice were maintained in specific-pathogen-free environment under a light/dark cycle of 12:12 h at a temperature of 23℃ and humidity of 50% to 70%; they had free access to food (chow diet) and water (14).

**(i) Experiment I: mouse small intestine microbiota collection.** Three-week-old ICR male mice were dosed with double-distilled water (ddH$_2$O) as the control or AOS per 10 mg/kg body weight (BW) via oral gavage (0.1 mL/mouse/day) (23, 24). The AOS dosing solution was freshly prepared every day and was delivered every morning for 3 weeks. There were two groups (30 mice/treatment), (i) control (ddH$_2$O), and (ii) A10 (AOS per 10 mg/kg BW). After 3 weeks treatment, the animals were maintained on regular diet for 3 more days (no treatment). Then, the mice were humanely euthanized to collect small intestinal luminal content (microbiota).

**(ii) Experiment II: high-fat diet (HFD) and streptozotocin (STZ) treatment and microbiota transplantation (FMT).** The small intestine luminal content (microbiota) from each group was pooled and homogenized, diluted 1:1 in 20% sterile glycerol (saline), and frozen (23, 24, 44, 45). Before inoculation, fecal samples were diluted in sterile saline to a working concentration of 0.05 g/mL and filtered through a 70-$\mu$m cell strainer. STZ was from Sigma (catalog no. S0130). Three-week-old ICR male mice were used in the current investigation. There were four treatment groups (30 mice/treatment), including (i) Con (control; regular diet plus saline); (ii) HFD+STZ (HS) (HFD from 3 to 10 weeks of age; at 5 weeks old, one dose of STZ at 85 mg/kg body weight after preliminary screening) (32, 33); (iii) Con-FMT (HFD from 3 to 10 weeks of age; at 5 weeks old, one dose of STZ at 85 mg/kg body weight; gut microbiota transplantation [FMT] from control mice [experiment I] from 5 to 7 weeks of age); and (iv) A10-FMT (HFD from 3 to 10 weeks of age; at 5-week-old one dose of STZ at 85 mg/kg body weight; FMT from AOS 10-mg/kg dosed mice [experiment I] from 5 to 7 weeks of age). HFD started from 3 weeks old (continued until the end of the experiment), and one dose of STZ was injected i.p. at 5 weeks old (2 weeks after HFD feeding) (32, 33). Then, the mice received oral FMT inoculations (0.1 mL) once daily for 2 weeks (5 weeks of age to 7 weeks of age) (23, 24). Then, the mice were regularly maintained (on respective diets) for another 3 weeks (10 weeks of age). Then, the mice were humanely euthanized to collect samples for different analyses (Fig. 1a, study scheme).

**Evaluation of spermatozoa motility using a computer-assisted sperm analysis system.** Spermatozoa motility was assessed using a computer-assisted sperm assay (CASA) method according to World Health Organization guidelines (46). After euthanasia, spermatozoa were collected from the cauda epididymis of mice and suspended in Dulbecco's modified Eagle medium (DMEM)-F-12 medium with 10% FBS and incubated at 37.5℃ for 30 min; samples were then placed in a prewarmed counting chamber. The Microptic sperm class analyzer (CASA system) was used in this investigation. It was equipped with a 20-fold objective, a camera adaptor (Eclipse E200; Nikon, Japan), and a camera (acA780-75gc; Basler, Germany), and it was operated by an SCA sperm class analyzer (Microptic S.L.). The classification of sperm motility was as follows: grade A linear velocity, >22 $\mu$m s$^{-1}$; grade B linear velocity, <22 $\mu$m s$^{-1}$; curvilinear velocity, >5 $\mu$m s$^{-1}$; grade C curvilinear velocity, <5 $\mu$m s$^{-1}$; and grade D, immotile spermatozoa. The spermatozoa motility data represented only grade A and grade B since only these two grades are considered to be functional.

**Morphological observations of spermatozoa.** The extracted murine caudal epididymides were placed in RPMI medium and finely chopped, and then, eosin Y (1%) was added for staining as described previously (46). Spermatozoan abnormalities were then viewed using an optical microscope and were classified into head or tail morphological abnormalities, two heads, two tails, blunt hooks, and short tails. The examinations were repeated three times, and 500 spermatozoa per animal were scored.

**RNA isolation and RNA-seq analyses.** Briefly, total RNA was isolated using TRIzol reagent (Invitrogen) and purified using a PureLink RNA minikit (catalog no. 12183018A; Life Technologies) following the manufacturers' protocol (46). Total RNA samples were first treated with DNase I to degrade any possible DNA contamination. Then, the mRNA was enriched using oligo(dT) magnetic beads. Mixed with the fragmentation buffer, the mRNA was broken into short fragments (about 200 bp), after which the first strand of cDNA was synthesized using a random-hexamer primer. Buffer, deoxynucleoside triphosphate (dNTP), RNase H, and DNA polymerase I were added to synthesize the second strand. The double-stranded cDNA was purified with magnetic beads. Subsequently, 3'-end single-nucleotide adenine (A) addition was performed. Finally, sequencing adaptors were ligated to the fragments. The fragments were enriched by PCR amplification. During the quality control (QC) step, an Agilent 2100 bioanalyzer and ABI StepOnePlus real-time PCR system were used to qualify and quantify the sample library. The library products were prepared for sequencing in an Illumina HiSeq 2500 system. The reads were mapped to

reference genes using SOAPaligner (v.2.20) with a maximum of two nucleotide mismatches allowed at the parameters of -m 0 -x 1000 -s 40 -l 35 -v 3 -r 2. The read number of each gene was transformed into reads per kilobases per million reads (RPKMs), and then differentially expressed genes were identified using the DEGseq package and the MA plot-based method with random sampling model (MARS) method. The threshold was set as false-discovery rate (FDR) of $\leq$0.001 and an absolute value of $\log_2$ ratio $\geq$1 to judge the significance of the difference in gene expression. Then, the data were analyzed by gene ontology (GO) and KEGG enrichment.

**Sequencing of microbiota from intestine digesta samples and data analysis. (i) DNA extraction.** Total genomic DNA of small intestine, cecum, and colon digesta was isolated using an E.Z.N.A. stool DNA kit (Omega Bio-tek, Inc., USA) following the manufacturer's instructions. DNA quantity and quality were analyzed using NanoDrop 2000 (Thermo Scientific, USA) and 1% agarose gel. Ten samples per group were determined.

**(ii) Library preparation and sequencing.** The V3-V4 region of the 16S rRNA gene was amplified using the primers MPRK341F (5'-ACTCCTACGGGAGGCAGCAG-3') and MPRK806R (5'-GGACTACHVGGG TWTCTAAT-3') with barcoding. The PCRs (total, 30 $\mu$L) included 15 $\mu$L Phusion high-fidelity PCR master mix (New England Biolabs), 0.2 mM primers, and 10 ng DNA. The thermal cycle was carried out with an initial denaturation at 98°C, followed by 30 cycles of 98°C for 10 s, 50°C for 30 s, 72°C for 30 s, and a final extension at 72°C for 5 min. PCR products were purified using a GeneJet gel extraction kit (Thermo Scientific, USA). The sequencing libraries were constructed with NEBNext Ultra DNA library prep kit for Illumina (NEB, USA) following the manufacturer's instructions, and index codes were added. Then, the library was sequenced on the Illumina HiSeq 2500 platform, and 300-bp paired-end reads were generated at the Novo gene. The paired-end reads were merged using FLASH (v.1.2.71). The quality of the tags was controlled in QIIME (v.1.7.02); meanwhile, all chimeras were removed. The "core set" of the Greengenes database was used for classification, and sequences with >97% similarity were assigned to the same operational taxonomic units (OTUs).

**(iii) Analysis of sequencing data.** Operational taxonomic unit abundance information was normalized using a standard sequence number corresponding to the sample with the least sequences. The alpha diversity index was calculated with QIIME (v.1.7.0). The UniFrac distance was obtained using QIIME (b1.7.0), and principal-coordinate analysis (PCoA) was performed using R software (v.2.15.3). The linear discriminate analysis effect size (LEfSe) was performed to determine differences in abundance; the threshold LDA score was 4.0. GraphPad Prism7 software was used to produce the graphs.

**Plasma and testis metabolite measurements by LC-MS/MS.** Plasma samples were collected and immediately stored at −80°C. Before LC-MS/MS analysis, the samples were thawed on ice and processed to remove proteins. Testis samples were collected, and the same amount of tissue from each mouse testis was used to isolate the metabolites using $CH_3OH$ to $H_2O$ (vol/vol) ratio of 4:1. Then samples were detected by Acquity UPLC and AB Sciex TripleTOF 5600 (LC/MS) as reported previously (23, 46). Ten samples per group were analyzed for plasma or testis samples. The HPLC conditions employed an Acquity UPLC ethylene-bridged hybrid (BEH) $C_{18}$ column (100 mm by 2.1 mm, 1.7 $\mu$m), solvent A (aqueous solution with 0.1% [vol/vol] formic acid), and solvent B (acetonitrile with 0.1% [vol/vol] formic acid) with a gradient program. The flow rate was 0.4 mL/min, and the injection volume was 5 $\mu$L. Progenesis QI v.2.3 (Nonlinear Dynamics, Newcastle, UK) was implemented to normalize the peaks. Then, the Human Metabolome Database (HMDB), LIPID MAPs (v.2.3), and METLIN software were used to qualify the data. Moreover, the data were processed with SIMCA software (v.14.0; Umetrics, Umeå, Sweden) following pathway enrichment analysis using the KEGG database (https://www.genome.jp/kegg/pathway.html).

**Determination of blood insulin, ALT, AST, TG, and T-AOC.** Blood insulin was determined by the kit from Beijing Solarbio Science & Technology Co., Ltd. (Beijing, People's Republic of China; catalog no. SEKM0141). Blood alanine aminotransferase (ALT; catalog no. C009-2-1), aspartate transaminase (AST; catalog no. C010-2-1), TG (catalog no. A110-1-1), and T-AOC (catalog no. A015-2-1) were determined by the kits from Nanjing Jiancheng Bioengineering Institute (Nanjing, People's Republic of China) (47). All procedures were followed from the manufacturers' instructions.

**Histopathological analysis.** Testicular tissues were fixed in 10% neutral buffered formalin, paraffin embedded, cut into 5-$\mu$m sections. and subsequently stained with hematoxylin and eosin (H&E) for histopathological analysis (46).

**Western blotting.** Western blotting of proteins was carried out as previously reported (23, 46). Briefly, testicular tissue samples were lysed in radioimmunoprecipitation assay (RIPA) buffer containing the protease inhibitor cocktail from Sangong Biotech, Ltd. (Shanghai, China). Protein concentration was determined using a bicinchoninic acid (BCA) kit (Beyotime Institute of Biotechnology, Shanghai, China). Goat anti-actin was used as a loading control. The information for primary antibodies (Abs) is listed in Table S1 in the supplemental material. Secondary donkey anti-goat Ab (catalog no. A0181) was purchased from Beyotime Institute of Biotechnology, and goat anti-rabbit (catalog no. A24531) Abs were bought from Novex by Life Technologies (USA). Fifty micrograms of total protein per sample was loaded onto 10% SDS-polyacrylamide electrophoresis gels. The gels were transferred to a polyvinylidene fluoride (PVDF) membrane at 300 mA for 2.5 h at 4°C. The membranes were then blocked with 5% bovine serum albumin (BSA) for 1 h at room temperature (RT), followed by three washes with 0.1% Tween 20 in TBS (TBST). The membranes were incubated with primary Abs diluted 1:500 in TBST with 1% BSA overnight at 4°C. After three washes with TBST, the blots were incubated with the horseradish peroxidase (HRP)-labeled secondary goat anti-rabbit or donkey anti-goat Ab, respectively, for 1 h at RT. After three washes, the blots were imaged. The bands were quantified using ImageJ software. The intensity of the specific protein band was normalized to actin first, and then the data were normalized to the control. The experiment was repeated >6 times.

**Detection of protein levels and location in testis using immunofluorescence staining.** The methodology for immunofluorescence staining of testicular samples is reported in our recent publications (23, 46). Sections of testicular tissue (5 $\mu$m) were prepared and subjected to antigen retrieval and immunostaining as previously described. Briefly, sections were first blocked with normal goat serum in phosphate-buffered saline (PBS), followed by incubation with primary Abs (1:100 in PBS-0.5% Triton X-100; Bioss Co. Ltd. Beijing, People's Republic of China) (Table S1) at 4°C overnight. After a brief wash, sections were incubated with an Alexa 546-labeled goat anti-rabbit secondary Ab (1:100 in PBS; Molecular Probes, Eugene, OR, USA) at RT for 30 min and then counterstained with 4′,6-diamidino-2-phenylindole (DAPI). The stained sections were examined using a Leica laser scanning confocal microscope (Leica TCS SP5 II; Germany). Ten animal samples from each treatment group were analyzed. Positively stained cells were counted. A minimum of 1,000 cells were counted for each sample of each experiment. The data were then normalized to the control.

**Statistical analysis.** Data were analyzed using SPSS statistical software (IBM Co., NY, USA) with one-way analysis of variance (ANOVA) with least significant difference (LSD) multiple comparison. The data were shown as the mean $\pm$ standard error of the mean (SEM). Statistical significance was based on $P$ values of $<0.05$.

**Ethical approval.** All animal procedures used in this study were approved by the Animal Care and Use Committee of the Institute of Animal Sciences of Chinese Academy of Agricultural Sciences.

**Data availability.** Liver RNA-seq raw data were deposited in NCBI's Gene Expression Omnibus under accession number GSE179098. The microbiota raw sequencing data generated in this study have been uploaded to the NCBI SRA database with the accession number PRJNA742204 (small intestine), PRJNA742202 (cecum), and PRJNA742203 (colon).

## SUPPLEMENTAL MATERIAL

Supplemental material is available online only.
**SUPPLEMENTAL FILE 1**, PDF file, 2.7 MB.
**SUPPLEMENTAL FILE 2**, XLSX file, 8.4 MB.
**SUPPLEMENTAL FILE 3**, XLSX file, 12.5 MB.

## ACKNOWLEDGMENTS

We thank the investigators and staff of the Beijing Genomics Institute (BGI) and Shanghai Luming Biotechnology Co., Ltd., for technical support.

This study was supported by the National Natural Science Foundation of China (32070536 and 31772408 to Y.Z.; 31672428 to H.Z.).

We declare no conflict of interest.

X.Y., Y.F., Y.H., R.Z., Y.J., X.T., D.L., and H.F. performed the experiments and analyzed the data. Y.Z., H.Z., L.C., and M.A. designed and supervised the study. Y.Z., X.T., and H.Z. wrote the manuscript. All the authors edited the manuscript and approved the final manuscript.

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
