## [Reviewer comments · Microbiology Spectrum]

Microbiology Spectrum

Gut-testis axis: microbiota prime metabolome to increase sperm quality in young type 2 diabetes

Xiaowei Yan, Yanni Feng, Yanan Hao, Ruqing Zhong, Yue Jiang, Xiangfang Tang, Dongxin Lu, Hanhan Fang, Manjree Agarwal, Liang Chen, Yong Zhao, and Hongfu Zhang

Corresponding Author(s): Yong Zhao, CAAS

Review Timeline:

Submission Date:	April 19, 2022
Editorial Decision:	May 31, 2022
Revision Received:	July 12, 2022
Editorial Decision:	August 9, 2022
Revision Received:	September 1, 2022
Accepted:	September 20, 2022

Editor: Henning Seedorf

Reviewer(s): The reviewers have opted to remain anonymous.

Transaction Report:

DOI: <https://doi.org/10.1128/spectrum.01423-22>

May 31, 2022

Dr. Yong Zhao
CAAS
2 Yuanmingyuan Western Road
Beijing
China

Re: Spectrum01423-22 (Gut-testis axis: microbiota prime metabolome to increase sperm quality in young type 2 diabetes)

Dear Dr. Yong Zhao:

Link Not Available

Sincerely,

Henning Seedorf

Journals Department
Reviewer comments:

Reviewer #1 (Comments for the Author):

Very happy to see such a collaborative effort which is why the five shared first authors is explained. The text would benefit from being edited. I started making edits but it soon became obvious that it would take too long within the review time window. I can however, do it if you wish so. It really is not much but the text would benefit at places. I have made some suggestions which might prove useful to you in the review document.

Reviewer #2 (Comments for the Author):

I thank the authors for performing this research on the influence of alginate oligosaccharide (AOS) supplemented gut microbiota on sperm quality in type 2 diabetes. I appreciate the effort put into the various experiments to get the results presented here. It seems a bit of care needs to be taken in the interpretation of some of the results of this research.

A few issues need to be addressed:

Lines 55-56: A10-FMT increased the gut 'Lactobacillus and Allobaculum'. However, Figure 1d (Allobaculum) shows no significant difference between the A10-FMT and Con-FMT. Similarly, Figure 1d (Lactobacillus) shows no significant differences across all groups. So, I think this cannot be attributed to the A10-FMT. Your results show that both genera were present in the groups tested though to different levels.

Figures 1a, b, c, and d are missing the 'Con' group. It would be nice to see what groups of bacteria were in this group on a normal diet and the relative amounts of the different genera along with the other groups. Though from Figure S1, looks like similar amounts of Allobaculum and Lactobacillus were present in the Con and HS groups

Lines 293-294: Here you are contradicting the results presented in Figure 1d on Allobaculum as there was no significant difference between the A10-FMT and Con-FMT groups. So, you will need to reword or get rid of the statement.

In general, from the results in Figure 1d, it will be erroneous to attribute the change in microbiota to Allobaculum and Lactobacillus alone as these genera were not exclusively present in this group only. I would suggest you look closely at your microbiota data to check if there are any differences attributable to specific genera for A10-FMT where it is significantly different from HS and Con-FMT. For example, members of the Ruminococcus genus are also short-chain fatty acid (SCFA) producers.

Lines 295-296: Looks like there is another contradiction here as Figure 3i shows similar levels of blood butyric acid in A10-FMT and Con-FMT. Was there a significant difference between the butyric acid from the A10-FMT vs Con-FMT groups? I assume not as there is no indication on the figure since you did indicate between the Con and HS groups. While I am not doubting the correlation between SCFA production and an increase in sperm concentration and motility, this must be rightly attributed to ensure reproducibility.

Figures 3d, h, l, and l: Important to compare A10-FMT vs Con-FMT to determine any significant differences. Please indicate using asterisks or ns

Line 67: Change 'benefited' to 'improved' in AOS-benefited gut microbiota (A10-FMT)

Line 105: Think 'as' is missing after 'such'

Line 111: Write R in Rhus coriaria in full as it appears once

Line 122. Please add a reference at the end of the sentence 'busulfan treated subjects'

Lines 140-141: Here you need to perform a comparison between the HS and the Con-FMT in Figure 1b before you can say it was not significantly different as you did for the glycogen in Figure 1c. This is because you indicated in Figure 1b, that there is no significant difference between A10-FMT vs Con-FMT.

Lines 149-150: Same issue as discussed previously with Allobaculum. Since the aim of this study is to show that AOS supplemented microbiota contributed to these findings, I wonder if it is not necessary to compare with the Con-FMT here. I mean, is it possible that the increase in Allobaculum is because of the FMT process and not because of the AOS since no significant differences in Allobaculum were observed in both groups (A10-FMT vs Con-FMT)?

Lines 229-233: Looks like no comparison was performed between the HS and Con-FMT here. For Figures 5a to 5p, I think it is important to see if any significant difference exists between these groups as shown in 5c and 5i. The reason behind this is because for some of these e.g., Figures 5e, m, o, and p, there were no significant differences between the A10-FMT and Con-FMT groups.

Line 718: Should be 'mmol/L' not mmol/L

Lines 727-739, 760-786. After writing the main legend, rather than repeating what the x and y axes are for each, you could write; For Figures, 3g, h, l, j, k, l, m, and n, the y-axis represents the relative amount and the x-axis represents the treatment, * $p < 0.05$ for all. This would save a lot of space and repetition as this is already shown in the Figures. The same applies to Figures 5c, d, e, f, g, h, l, l, m, n, o, and p.

Similarly, legends for Figures 6a, b, and c (Lines 789-794) and lines 804-809 for Figures 7b, c and d.

Inconsistency in the reference style, some have issue numbers and others do not. For example, references 23 and 24 (lines 612 and 615) are from the same journal but ref 23 is missing the issue number (1)

General considerations

It would be useful to know what statistical test was used for each of the figures, that is, for example, which of the data sets was the T-test used for? Since most of the figures are compared across three or four groups and appropriate tests with corrections for multiple comparisons are ideal. Though it is stated in the methods section, it is not clear which tests were used for which analysis. Consider adding, 'one-way ANOVA was used with LSD for multiple comparisons'. I wonder what the actual p values were as I suppose this was obtained during the analysis as the differences between some assays are different from others e.g

Figure 5A on DHA, the statistical difference between HS and A10-FMT was it less than $p < 0.01$ or $p < 0.05$?
The number of the figure ought to be indicated on each figure to make it easier to navigate since the figure legends are elsewhere.

Here is a paper on Allobaculum and its potential role in obesity: [Frontiers | Allobaculum Involves in the Modulation of Intestinal ANGPTL4 Expression in Mice Treated by High-Fat Diet | Nutrition \(frontiersin.org\)](https://doi.org/10.3389/fnut.2019.00011)

You may also want to look at this: [Identification of Allobaculum mucolyticum as a novel human intestinal mucin degrader - PMC \(nih.gov\)](https://pubmed.ncbi.nlm.nih.gov/31111111/)

Reviewer #3 (Comments for the Author):

The work "Gut-testis axis: microbiota prime metabolome to increase sperm quality in young type 2 diabetes", conducted by Dr. Xiaowei Yan and colleagues, enriches the field of male infertility and microbiota. The authors were able to respond elegantly and robustly to the issues that motivated the study. Using a murine model of type 1 diabetes, the authors were able to demonstrate that the transplantation of alginate oligosaccharide (AOS)-modified gut microbiota was able to significantly decreased blood glucose and recover 5 to 10 times the concentration and motility of the subjects' sperm.

Staff Comments:

Preparing Revision Guidelines

Please return the manuscript within 60 days; if you cannot complete the modification within this time period, please contact me. If you do not wish to modify the manuscript and prefer to submit it to another journal, please notify me of your decision immediately so that the manuscript may be formally withdrawn from consideration by Microbiology Spectrum.

Review manuscript “Gut-testis axis: microbiota prime metabolome to increase sperm quality in young type 2 diabetes”

Summary

The authors want to test the effect of AOS-gut modified microbiota by means of fecal microbiota transplantation on a type 2 diabetes phenotype in mice (A10-FMT). They induced a T2D phenotype by a high fat diet and injection of streptozotocin. They then treated mice with A10-FMT. They compared these to control animals, animals with an untreated T2D phenotype and animals with T2D but treated with control FMT (Con-FMT).

They determined a T2D phenotype by measuring blood glucose.

They assessed the effects of the different types of FMT by examining microbiota in different parts of the intestine, sperm quality, liver function, testicular metabolome and microenvironment, and spermatogenesis.

They claim that:

1. A10-FMT significantly decreased blood glucose
2. T2D shows sperm concentration and motility decrease
3. A10-FMT improved liver functions and systemic anti-oxidant environment
4. A10-FMT promoted spermatogenesis through the improvement in the expression of proteins important for spermatogenesis to increase sperm concentration and motility

The suggest that the potential mechanism is: “A10-FMT increased gut beneficial microbes *Lactobacillus* and *Allobaculum* to elevate blood and/or testicular butyric acid, DHA, EPA, and testosterone to promote spermatogenesis and thus to ameliorate sperm concentration and motility”.

Throughout the results sections they make summarizing claims of what their data supports, which are partly correct and partly not. In the following sections I make suggestions on how this can be improved so that the true changes observed are highlighted.

Major items

Test effectivity of T2D treatment with FMT on male fertility by performing fertility tests such as examining how many pregnancies come to term and offspring health status. This is particularly important as the authors state that T2D is linked to various symptoms of male infertility, not confined to sperm count and motility.

SupFig 4a-b are unsupported by the data. There is no change in either AST ALT by introducing the T2D phenotype so any changes with the treatment and likely not related.

Fig 2a-c – gut microbiota of control mice data are missing

Across the figures, very few data points appear at times. For example, FigS1, S4.

Fig S4b: Provide stats between the control groups and A10-FMT group. Where ALT is (significantly?) decreased in A10-FMT samples compared to Con and discuss that this could be a sign for vitB deficiency or chronic kidney disease.

Line 200: AST was “increased” in HS; it is not significant and could be just a trend. Provide stats. By the same means one could state that HS has decreased ALT compared to control.

FigS4f- state is H&E and must be corrected in text line 211: “The protein level of NR1H3 was increased in HS while it was decreased in A10-FMT but not in Con-FMT (Fig. 4a, f), which is consistent with the histopathology of the liver samples in different groups (Fig. S4f).” Fig S4f does not show a decrease in NR1H3. Please describe what about these sections is consistent with your findings.

Fig1c stats between control and HS groups need to be presented. If the difference is not significant then the FMT treatment does have an effect on the HS model, but it does not represent T2D. Statements always have to related to the wild-type.

Fig 3a stats. If there is no significance, then the claims in the text need to be re-written to accommodate for that (paragraph beginning at 180).

Fig3b stats between control and A10-FMT being non-significant would strengthen the statements made.

Fig 3D stats between control and HS is needed here again. The reason is that one can establish the phenotype of T2D and therefore the effects of any treatment on T2D ONLY when there is significance between control and HS (untreated T2D model).

Line 191 “Similarly, blood retinoic acid was decreased in HS, however, it was increased in A10-FMT (Fig. 3m).” needs to be rephrased to reflect the lack of significance. One can state trends and then in Fig4 where the protein synthesis of retinoic acid is discussed, remind the reader of Fig 3m.

Fig 3n It is not clear to me how these data were collected, and the conclusions made. Which group was used to make this analysis? Was it control to the HS groups? This information needs to be clear in the text and in the figure legend.

Fig 4J stats need to be provided for the control group. All inferences have to be made on the control regarding the presence of the phenotype. Fig 4i does not show significant increase in GPX1 between the HS and A10-FMT groups, if I am interpreting the “ab” labelling correctly. The particular choice of statistical strength labeling is confusing. The graphs can be broken down to simple graphs with data points, error bars and asterisks for significance.

Fig5 stats need to be provided for the control group throughout this figure or the inferences made are not as telling as stated in the paper. Figures 5d, e, h, l & m can be used to infer effectivity of the treatment. A potential treatment cannot just reflect a difference to the pathology but has to reflect similarity to the WT because there can also be adverse effects and side effects to each treatment which need to be made known for consideration.

Section “**A10-FMT improved the testicular metabolome and the testicular microenvironment**” needs to be re-written to reflect the effects between wt and HS groups as well. For example, the paper states that retinoids are important for spermatogenesis and that those are elevated in A10-FMT, but they are much higher than the wildtype and more importantly the HS group does not seem

to have a phenotype of reduced retinoids which signals that the specific measurement is not relevant to T2D.

The statement “HS treatment significantly decreased these hormones while A10-FMT led to a recovery (Fig. 6a-c).” is not true for Fig6c.

Fig 6f-g, again, it is not clear to me how these data were collected, and the conclusions made. Which group was used to make this analysis? Was it control to the HS groups? This information needs to be clear in the text and in the figure legend.

Figs 1d and 7d are conflicting with respect to the con-FMT data. Please discuss.

Fig 1f No significance between control and HS for TP1 and PIWIL1. Please rephrase in the text (lines 257-8). I would also recommend trying tubulin as a control because it seems that actin was overloaded and may not have served as an adequate loading control.

Minor items

Line 46: edit to something like “Young type diabetes 2 affects x% of population with a noted increase in cases in recent years”.

Line 47: “aimed at exploring”

Line 47: “beneficial improvements” is redundant. Remove “beneficial” or write “beneficial effects”

Replace “youth mice” with “young mice”. “of 5 weeks of age”.

Line 98 - inconsistent

Line 99 – references needed.

Line 148 reference

Line 186 reference

Line 216 reference.

Line 207 state which gene. “Expression of the lipid metabolism related gene was”

Line 229 reference

Line 234 reference

Sample numbers are not labelled in the figure legends nor at statistical tests used.

Include catalogue number if the AOS solution and streptozotocin used. I would have liked to look at the content of AOS.

Would be good to explain why the measurement of blood glucose level is enough to determine a model of T2D has been generated. Why HbA1C% test was not done, what is the speed of induced type1D development in mice differs to that of T2D. They could for example show that in their model T2D mice do express insulin, a key difference between T2D and T1D.

Line 367: state where streptozotocin was injected (ie. IP, IM, IV?).

Line 134 & 136: add the use of streptozotocin for a complete description of the study group of the FMT treated groups

Include Con-FMT schematic in Figure 1.

FigS1a – why is BW reduced in High fat diet mouse groups?

Fig S1b – what do point represent? 30 mice are stated per group but only 4-5 points appear on the graphs.

Table S1 – protein size for TP1 is 6.4kDa

Line 226 “Since A10-FMT improved liver functions and the blood metabolome, next we set out to explore the beneficial advantages of A10-FMT on the testicular metabolome”. The sentence is leading by stating beneficial effects on the testicular metabolome prior to showing the supportive data. I would remove the term beneficial at this point and welcome the reader to decide as the results are presented.

Line 256, stats not significant for SYCP3 so the statement is unsupported.

Based on Fig 7 con-FMT treatment seems to reverse the sperm quality problems and would resolve the specific part of the phenotype.

In the material and methods, perhaps include a statement as to why male mice were used and not females (hormone fluctuations etc).

The material and methods include a section on acrosome integrity which was not talked about in the paper or figures.

Line 512, how many sections were analysed?

It might interesting to discuss how long the effects of FMT treatment would last.

Response to reviewer's comments

First, we would like to thank the editors and reviewers very much for the comments.

Reviewer(s)' Comments to Author:

Reviewer #1 (Comments for the Author):

Very happy to see such a collaborative effort which is why the five shared first authors is explained. The text would benefit from being edited. I started making edits but it soon became obvious that it would take too long within the review time window. I can however, do it if you wish so. It really is not much but the text would benefit at places. I have made some suggestions which might prove useful to you in the review document.

Attachment:

Review manuscript "Gut-testis axis: microbiota prime metabolome to increase sperm quality in young type 2 diabetes"

Summary

The authors want to test the effect of AOS-gut modified microbiota by means of fecal microbiota transplantation on a type 2 diabetes phenotype in mice (A10-FMT). They induced a T2D phenotype by a high fat diet and injection of streptozotocin. They then treated mice with A10-FMT. They compared these to control animals, animals with an untreated T2D phenotype and animals with T2D but treated with control FMT (Con-FMT).

They determined a T2D phenotype by measuring blood glucose.

They assessed the effects of the different types of FMT by examining microbiota in different parts of the intestine, sperm quality, liver function, testicular metabolome and microenvironment, and spermatogenesis.

They claim that:

1. A10-FMT significantly decreased blood glucose
2. T2D shows sperm concentration and motility decrease
3. A10-FMT improved liver functions and systemic anti-oxidant environment
4. A10-FMT promoted spermatogenesis through the improvement in the expression of proteins important for spermatogenesis to increase sperm concentration and motility

The suggest that the potential mechanism is: “A10-FMT increased gut beneficial microbes *Lactobacillus* and *Allobaculum* to elevate blood and/or testicular butyric acid, DHA, EPA, and testosterone to promote spermatogenesis and thus to ameliorate sperm concentration and motility”.

Throughout the results sections they make summarizing claims of what their data supports, which are partly correct and partly not. In the following sections I make suggestions on how this can be improved so that the true changes observed are highlighted.

Major items

Comment #1: Test effectivity of T2D treatment with FMT on male fertility by performing fertility tests such as examining how many pregnancies come to term and offspring health status. This is particularly important as the authors state that T2D is linked to various symptoms of male infertility, not confined to sperm count and motility.

Response: Thank the reviewer very much for the very nice comment. In current study, we did not detect the pregnant rate or number of living pups/litter. We plan to perform experiments to determine it in the future. Thanks.

Comment #2: SupFig 4a-b are unsupported by the data. There is no change in either AST ALT by introducing the T2D phenotype so any changes with the treatment and likely not related.

Response: Thank the reviewer very much for the very nice comment. Since, the data are not significant between Con and HS for ALT and AST, FigS 4a-b were deleted as the reviewer suggested. Thanks.

Comment #3: Fig 2a-c – gut microbiota of control mice data are missing

Response: Thank the reviewer very much for the very nice comment.

Comment #4: Across the figures, very few data points appear at times. For example, FigS1, S4.

Response: Thank the reviewer very much for the very nice comment. All the analyses were done with more than three samples per group. And the metabolic analysis was done with 10 samples per group. Thanks.

Comment #5: Fig S4b: Provide stats between the control groups and A10-FMT group. Where ALT is (significantly?) decreased in A10-FMT samples compared to Con and discuss that this could be a sign for vitB deficiency or chronic kidney disease.

Response: Thank the reviewer very much for the very nice comment. Since the data are not significant between Con and HS for ALT and AST. FigS4l and b were deleted. Thanks.

Comment #6: Line 200: AST was “increased” in HS; it is not significant and could be just a trend. Provide stats. By the same means one could state that HS has decreased ALT compared to control.

Response: Thank the reviewer very much for the very nice comment. Since the data are not significant between Con and HS for ALT and AST. FigS4l and b were deleted. We also revised the manuscript. Thanks.

Comment #7: FigS4f- state is H&E and must be corrected in text line 211: “The protein level of NR1H3 was increased in HS while it was decreased in A10-FMT but not in Con-FMT (Fig. 4a, f), which is consistent with the histopathology of the liver samples in different groups (Fig. S4f).”. Fig S4f does not show a decrease in NR1H3. Please describe what about these sections is consistent with your findings.

Response: Thank the reviewer very much for the very nice comment. We are sorry this section confused the reviewer. It was revised as the reviewer suggested. Thanks.

Comment #8: Fig1c stats between control and HS groups need to be presented. If the difference is not significant then the FMT treatment does have an effect on the HS model, but it does not represent T2D. Statements always have to related to the wild-type.

Response: Thank the reviewer very much for the very nice comment. The difference is not significant between Con and HS group. We added stats for Fig. 1c and revised the text. Thanks.

Comment #9: Fig 3a stats. If there is no significance, then the claims in the text need to be re-written to accommodate for that (paragraph beginning at 180).

Response: Thank the reviewer very much for the very nice comment. We added stats for Fig. 1c and revised the text as the reviewer suggested. Thanks.

Comment #10: Fig3b stats between control land A10-FMT being non-significant would strengthen the statements made.

Response: Thank the reviewer very much for the very nice comment. Yes, there is no significant difference between Con and A10-FMT in Fig. 3B. Thanks.

Comment #11: Fig 3D stats between control and HS is needed here again. The reason is that one can establish the phenotype of T2D and therefore the effects of any treatment on T2D ONLY when there is significance between control and HS (untreated T2D model).

Response: Thank the reviewer very much for the very nice comment. We added stats for Fig. 1c and revised the text as the reviewer suggested. Thanks.

Comment #12: Line 191 “Similarly, blood retinoic acid was decreased in HS, however, it was increased in A10-FMT (Fig. 3m).” needs to be rephrased to reflect the lack of significance. One can state trends and then in Fig4 where the protein synthesis of retinoic acid is discussed, remind the reader of Fig 3m.

Response: Thank the reviewer very much for the very nice comment. We revised the text as the reviewer suggested. Thanks.

Comment #13: Fig 3n It is not clear to me how these data were collected, and the

conclusions made. Which group was used to make this analysis? Was it control to the HS groups? This information needs to be clear in the text and in the figure legend.

Response: Thank the reviewer very much for the very nice comment. We are sorry. This is the correlation analysis between blood metabolites and sperm motility/concentration. All the data were used to do the analysis. The information has been added into the legend. Thanks.

Comment #14: Fig 4J stats need to be provided for the control group. All inferences have to be made on the control regarding the presence of the phenotype. Fig 4i does not show significant increase in GPX1 between the HS and A10-FMT groups, if I am interpreting the “ab” labelling correctly. The particular choice of statistical strength labeling is confusing. The graphs can be broken down to simple graphs with data points, error bars and asterisks for significance.

Response: Thank the reviewer very much for the very nice comment. We added stats for Fig. 4j and revised the text as the reviewer suggested. Yes, the data were significant between Con and HS for GPX1. We revised the text. We have tried to make the graph clear for Fig. 4 i, however, if all the data shown in one graph, it is hard to make it very simple. Thank the review for the understanding. Thanks.

Comment #15: Fig5 stats need to be provided for the control group throughout this figure or the inferences made are not as telling as stated in the paper. Figures 5d, e, h, l & m can be used to infer effectivity of the treatment. A potential treatment cannot just reflect a difference to the pathology but has to reflect similarity to the WT because there can also be adverse effects and side effects to each treatment which need to be made known for consideration.

Response: Thank the reviewer very much for the very nice comment. We added stats for Fig. 5 as the reviewer suggested, and revised the text as the reviewer suggested. We understand what the reviewer concerned. We are trying to make the result clear and correct to the readers as the reviewer suggested. Thanks.

Comment #16: Section “A10-FMT improved the testicular metabolome and the testicular microenvironment” needs to be re-written to reflect the effects between

wt and HS groups as well. For example, the paper states that retinoids are important for spermatogenesis and that those are elevated in A10-FMT, but they are much higher than the wildtype and more importantly the HS group does not seem to have a phenotype of reduced retinoids which signals that the specific measurement is not relevant to T2D.

Response: Thank the reviewer very much for the very nice comment. We revised this paragraph as the reviewer suggested. Thanks.

Comment #17: The statement “HS treatment significantly decreased these hormones while A10-FMT led to a recovery (Fig. 6a-c).” is not true for Fig6c.

Response: Thank the reviewer very much for the very nice comment. It was revised as the reviewer suggested. Thanks.

Comment #18: Fig 6f-g, again, it is not clear to me how these data were collected, and the conclusions made. Which group was used to make this analysis? Was it control to the HS groups? This information needs to be clear in the text and in the figure legend.

Response: Thank the reviewer very much for the very nice comment. They are the correlation analysis. All the testis metabolites and sperm motility/concentration in all the groups were analyzed for Fig 6f. All the testis metabolites and all the blood metabolites in all the groups were analyzed for Fig 6g. Thanks.

Comment #19: Figs 1d and 7d are conflicting with respect to the con-FMT data. Please discuss.

Response: Thank the reviewer very much for the very nice comment. In Fig. 1d sperm motility, the data in Con-FMT and Con were similar (no difference). IN Fig. 7d PGK2, although it is a little higher in Con-FMT group, there is no difference for Con-FMT and Con. Thanks.

Comment #20: Fig 7f No significance between control and HS for TP1 and PIWIL1.

Please rephrase in the text (lines 257-8). I would also recommend trying tubulin as a control because it is seems that actin was overloaded and may not have served as an adequate loading control.

Response: Thank the reviewer very much for the very nice comment. It was revised as the reviewer suggested. We have also tried GAPDH as the loading control. And the data were similar. Thanks.

Minor items

Comment #21: Line 46: edit to something like “Young type diabetes 2 affects x% of population with a noted increase in cases in recent years”.

Response: Thank the reviewer very much for the very nice comment. It was revised as the reviewer suggested. Thanks.

Comment #22: Line 47: “aimed at exploring”

Response: Thank the reviewer very much for the very nice comment. It was revised as the reviewer suggested. Thanks.

Comment #23: Line 47: “beneficial improvements” is redundant. Remove “beneficial” or write “beneficial effects”

Response: Thank the reviewer very much for the very nice comment. It was revised as the reviewer suggested. Thanks.

Comment #24: Replace “youth mice” with “young mice”. “of 5 weeks of age”.

Response: Thank the reviewer very much for the very nice comment. It was revised as the reviewer suggested. Thanks.

Comment #25: Line 98 - inconsistent

Response: Thank the reviewer very much for the very nice comment. It was revised

as the reviewer suggested. Thanks.

Comment #26: Line 99 – references needed.

Response: Thank the reviewer very much for the very nice comment. The references have been added as the reviewer suggested. Thanks.

Comment #27: Line 148 reference

Response: Thank the reviewer very much for the very nice comment. The references have been added as the reviewer suggested. Thanks.

Comment #28: Line 186 reference

Response: Thank the reviewer very much for the very nice comment. The references have been added as the reviewer suggested. Thanks.

Comment #29: Line 216 reference.

Response: Thank the reviewer very much for the very nice comment. The references have been added as the reviewer suggested. Thanks.

Comment #30: Line 207 state which gene. “Expression of the lipid metabolism related gene was”

Response: Thank the reviewer very much for the very nice comment. There were a group of genes, some examples have been added. Thanks.

Comment #31: Line 229 reference

Response: Thank the reviewer very much for the very nice comment. The references have been added as the reviewer suggested. Thanks.

Comment #32: Line 234 reference

Response: Thank the reviewer very much for the very nice comment. The references have been added as the reviewer suggested. Thanks.

Comment #33: Sample numbers are not labelled in the figure legends nor at statistical tests used.

Response: Thank the reviewer very much for the very nice comment. The sample numbers have been added in the Figure legends. Thanks.

Comment #34: Include catalogue number if the AOS solution and streptozotocin used. I would have liked to look at the content of AOS.

Would be good to explain why the measurement of blood glucose level is enough to determine a model of T2D has been generated. Why HbA1C% test was not done, what is the speed of induced type1D development in mice differs to that of T2D. They could for example show that in their model T2D mice do express insulin, a key difference between T2D and T1D.

Response: Thank the reviewer very much for the very nice comment. The catalogue number for streptozotocin was added in the Method Section. AOS are composed of α -L-gulonate (G) and β -D-mannuronate (M) joined by 1, 4-glycoside bonds. There were 2, 3, 4, 5 and 6 residues of G and/or M formed oligosaccharides for AOS in current study. We are sorry. We just determined blood glucose as reported in many articles. We will determine HbA1C in our future studies. Three days after injection of STZ, the blood glucose was increased to high enough to be diabetic as suggested in many articles. And insulin was detected in current study (Fig. S1b). It was decreased by HS. However, A10-FMT or Con-FMT did not changed it compared to HS. Thanks.

Comment #35: Line 367: state where streptozotocin was injected (ie. IP, IM, IV?).

Response: Thank the reviewer very much for the very nice comment. The information has been added as the reviewer suggested. Thanks.

Comment #36: Line 134 & 136: add the use of streptozotocin for a complete

description of the study group of the FMT treated groups

Response: Thank the reviewer very much for the very nice comment. The information has been added as the reviewer suggested. Thanks.

Comment #37: Include Con-FMT schematic in Figure 1.

Response: Thank the reviewer very much for the very nice comment. It was revised as the reviewer suggested. Thanks.

Comment #38: FigS1a – why is BW reduced in High fat diet mouse groups?

Response: Thank the reviewer very much for the very nice comment. At the beginning we also thought it strange. We repeated it one more time, the result was the same. This may be because STZ treatment. In HFD treatment only (no STZ), the body weight was higher than Control group. Thanks.

Comment #39: Fig S1b – what do point represent? 30 mice are stated per group but only 4-5 points appear on the graphs.

Response: Thank the reviewer very much for the very nice comment. Each point represents one sample. For insulin, the student did 4 samples per group. Thanks.

Comment #40: Table S1 – protein size for TP1 is 6.4kDa

Response: Thank the reviewer very much for the very nice comment. The information has been added. Thank the reviewer very much.

Comment #41: Line 226 “Since A10-FMT improved liver functions and the blood metabolome, next we set out to explore the beneficial advantages of A10-FMT on the testicular metabolome”. The sentence is leading by stating beneficial effects on the testicular metabolome prior to showing the supportive data. I would remove the term beneficial at this point and welcome the reader to decide as the results are presented.

Response: Thank the reviewer very much for the very nice comment. It was revised as the reviewer suggested. Thanks.

Comment #42: Line 256, stats not significant for SYCP3 so the statement is unsupported.

Response: Thank the reviewer very much for the very nice comment. It was revised as the reviewer suggested. Thanks.

Comment #43: Based on Fig 7 con-FMT treatment seems to reverse the sperm quality problems and would resolve the specific part of the phenotype.

Response: Thank the reviewer very much for the very nice comment. Con-FMT produced some effect, however, the data were not so profound as A10-FMT. Thanks.

Comment #44: In the material and methods, perhaps include a statement as to why male mice were used and not females (hormone fluctuations etc).

Response: Thank the reviewer very much for the very nice comment. The information has been added as the reviewer suggested. Thanks.

Comment #45: The material and methods include a section on acrosome integrity which was not talked about in the paper or figures.

Response: Thank the reviewer very much for the very nice comment. The information in methods section has been revised as the reviewer suggested. Thanks.

Comment #46: Line 512, how many sections were analysed?

Response: Thank the reviewer very much for the very nice comment. At least three different sections have been analyzed. Thanks.

Comment #47: It might interesting to discuss how long the effects of FMT treatment would last.

Response: Thank the reviewer very much for the very nice comment. In another study, we found the effect of FMT can last more than one year if the animals maintained in the same condition (no other treatment). Thanks.

Reviewer #2 (Comments for the Author):

I thank the authors for performing this research on the influence of alginate oligosaccharide (AOS) supplemented gut microbiota on sperm quality in type 2 diabetes. I appreciate the effort put into the various experiments to get the results presented here. It seems a bit of care needs to be taken in the interpretation of some of the results of this research.

A few issues need to be addressed:

Comment #1: Lines 55-56: A10-FMT increased the gut 'Lactobacillus and Allobaculum'. However, Figure 1d (Allobaculum) shows no significant difference between the A10-FMT and Con-FMT. Similarly, Figure 1d (Lactobacillus) shows no significant differences across all groups. So, I think this cannot be attributed to the A10-FMT. Your results show that both genera were present in the groups tested though to different levels.

Response: Thank the reviewer very much for the very nice comment. In small intestine, compared to HS group A10-FMT increased Allobaculum significantly, however Con-FMT increased Allobaculum in some extent (not significant). And there was no significant difference between A10-FMT and Con-FMT.

In small intestine, A10-FMT and Con-FMT similarly increased Lactobacillus in some extent (not significant) compared to HS group.

In cecum, compared to HS group, A10-FMT and Con-FMT increased Lactobacillus in some extent (not significant), while it was slightly higher in A10-FMT than Con-FMT (not significant). We revised the abstract as the reviewer suggested. Thanks.

Comment #2: Figures 1a, b, c, and d are missing the 'Con' group. It would be nice to see what groups of bacteria were in this group on a normal diet and the relative

amounts of the different genera along with the other groups. Though from Figure S1, looks like similar amounts of Allobaculum and Lactobacillus were present in the Con and HS groups

Response: Thank the reviewer very much for the very nice comment. The comparison of bacteria for Con group and HS group (small intestine, cecum, colon) are present in Fig. S1e, f, i, j, m, n. In order to make the data clear for A10-FMT and Con-FMT compared to HS group, the data for Con group were not added in Fig 2a, b, c and d (the reviewer mentioned as Fig. 1a, b, c, and d). Yes, the amounts of Allobaculum and Lactobacillus were similar in the Con and HS groups. Helicobacter was increased in cecum and colon, which was decreased by A10-FMT and/or Con-FMT. Thanks.

Comment # 3: Lines 293-294: Here you are contradicting the results presented in Figure 1d on Allobaculum as there was no significant difference between the A10-FMT and Con-FMT groups. So, you will need to reword or get rid of the statement.

Response: Thank the reviewer very much for the very nice comment. We have revised it as the reviewer suggested. Thanks.

Comment # 4: In general, from the results in Figure 1d, it will be erroneous to attribute the change in microbiota to Allobaculum and Lactobacillus alone as these genera were not exclusively present in this group only. I would suggest you look closely at your microbiota data to check if there are any differences attributable to specific genera for A10-FMT where it is significantly different from HS and Con-FMT. For example, members of the Ruminococcus genus are also short-chain fatty acid (SCFA) producers.

Response: Thank the reviewer very much for the very nice comment. As we responded in Comment #1, there are a few microbes were changed by A10-FMT significantly while altered by Con-FMT not significantly. The effects of A10-FMT may be multiple effects from a few different microbes. We revised the manuscript to make it more clearly. Thanks.

Comment # 5: Lines 295-296: Looks like there is another contradiction here as Figure 3i shows similar levels of blood butyric acid in A10-FMT and Con-FMT. Was there a significant difference between the butyric acid from the A10-FMT vs Con-FMT groups? I assume not as there is no indication on the figure since you did indicate between the Con and HS groups. While I am not doubting the correlation between SCFA production and an increase in sperm concentration and motility, this must be rightly attributed to ensure reproducibility.

Response: Thank the reviewer very much for the very nice comment. Yes, although the level of butyric acid was higher in A10-FMT, the difference was not significant between A10-FMT and Con-FMT. We revised the text as the reviewer suggested. Thanks.

Comment # 6: Figures 3d, h, I, and l: Important to compare A10-FMT vs Con-FMT to determine any significant differences. Please indicate using asterisks or ns

Response: Thank the reviewer very much for the very nice comment. We revised as the reviewer suggested. And the text was revised. Thanks.

Comment # 7: Line 67: Change 'benefited' to 'improved' in AOS-benefited gut microbiota (A10-FMT)

Response: Thank the reviewer very much for the very nice comment. It was revised as the reviewer suggested. Thanks.

Comment # 8: Line 105: Think 'as' is missing after 'such'

Response: Thank the reviewer very much for the very nice comment. It was added as the reviewer suggested. Thanks.

Comment # 9: Line 111: Write R in Rhus coriaria in full as it appears once

Response: Thank the reviewer very much for the very nice comment. It was added as the reviewer suggested. Thanks.

Comment # 10: Line 122. Please add a reference at the end of the sentence 'busulfan treated subjects'

Response: Thank the reviewer very much for the very nice comment. The references have been added as the reviewer suggested. Thanks.

Comment # 11: Lines 140-141: Here you need to perform a comparison between the HS and the Con-FMT in Figure 1b before you can say it was not significantly different as you did for the glycogen in Figure 1c. This is because you indicated in Figure 1b, that there is no significant difference between A10-FMT vs Con-FMT.

Response: Thank the reviewer very much for the very nice comment. Yes, it was not significantly different between HS and Con-FMT. We revised the Fig 1 as the reviewer suggested. Thanks.

Comment # 12: Lines 149-150: Same issue as discussed previously with Allobaculum. Since the aim of this study is to show that AOS supplemented microbiota contributed to these findings, I wonder if it is not necessary to compare with the Con-FMT here. I mean, is it possible that the increase in Allobaculum is because of the FMT process and not because of the AOS since no significant differences in Allobaculum were observed in both groups (A10-FMT vs Con-FMT)?

Response: Thank the reviewer very much for the very nice comment. We understand the reviewer said. In small intestine, compared to HS group A10-FMT increased Allobaculum significantly, however Con-FMT increased Allobaculum in some extent (not significant). And there was no significant difference between A10-FMT and Con-FMT. The effects of A10-FMT may be multiple effects from a few different microbes. Thanks.

Comment # 13: Lines 229-233: Looks like no comparison was performed between the HS and Con-FMT here. For Figures 5a to 5p, I think it is important to see if any significant difference exists between these groups as shown in 5c and 5i. The reason behind this is because for some of these e.g., Figures 5e, m, o, and p, there were no

significant differences between the A10-FMT and Con-FMT groups.

Response: Thank the reviewer very much for the very nice comment. We added the statistic analysis into Fig. 5 as the reviewer suggested, and revised the text as the reviewer suggested. Thanks.

Comment # 14: Line 718: Should be 'mmol/L' not mmlo/L

Response: Thank the reviewer very much for the very nice comment. It was revised. Thanks.

Comment # 15: Lines 727-739, 760-786. After writing the main legend, rather than repeating what the x and y axes are for each, you could write; For Figures, 3g, h, I, j, k, l, m, and n, the y-axis represents the relative amount and the x-axis represents the treatment, *p < 0.05 for all. This would save a lot of space and repetition as this is already shown in the Figures. The same applies to Figures 5c, d, e, f, g, h, I, l, m, n, o, and p. Similarly, legends for Figures 6a, b, and c (Lines 789-794) and lines 804-809 for Figures 7b, c and d.

Response: Thank the reviewer very much for the very nice comment. We have revised it as the reviewer suggested. Thank the reviewer very much.

Comment # 16: Inconsistency in the reference style, some have issue numbers and others do not. For example, references 23 and 24 (lines 612 and 615) are from the same journal but ref 23 is missing the issue number (1)

Response: Thank the reviewer very much for the very nice comment. We have revised it as the reviewer suggested. Thanks.

General considerations

Comment # 17: It would be useful to know what statistical test was used for each of the figures, that is, for example, which of the data sets was the T-test used for? Since most of the figures are compared across three or four groups and appropriate tests with corrections for multiple comparisons are ideal. Though it is stated in the methods

section, it is not clear which tests were used for which analysis. Consider adding, 'one-way ANOVA was used with LSD for multiple comparisons'. I wonder what the actual p values were as I suppose this was obtained during the analysis as the differences between some assays are different from others e.g Figure 5A on DHA, the statistical difference between HS and A10-FMT was it less than $p < 0.01$ or $p < 0.05$?

Response: Thank the reviewer very much for the very nice comment. The statistic analysis was ANOVA. We revised the method section. And all the significance was at $p < 0.05$. Thanks.

Comment # 18: The number of the figure ought to be indicated on each figure to make it easier to navigate since the figure legends are elsewhere.

Response: Thank the reviewer very much for the very nice comment. We added figure number on the figures as the reviewer suggested. Thanks.

Comment # 19: Here is a paper on Allobaculum and its potential role in obesity: Frontiers | Allobaculum Involves in the Modulation of Intestinal ANGPTL4 Expression in Mice Treated by High-Fat Diet | Nutrition (frontiersin.org)

Response: Thank the reviewer very much. The article is important for us. Thanks.

Comment # 20: You may also want to look at this: Identification of Allobaculum mucolyticum as a novel human intestinal mucin degrader - PMC (nih.gov)

Response: Thank the reviewer very much. The article is important for us. Thanks.

Reviewer #3 (Comments for the Author):

The work "Gut-testis axis: microbiota prime metabolome to increase sperm quality in young type 2 diabetes", conducted by Dr. Xiaowei Yan and colleagues, enriches the field of male infertility and microbiota. The authors were able to respond elegantly and robustly to the issues that motivated the study. Using a murine model of type 1 diabetes, the authors were able to demonstrate that the transplantation of alginate oligosaccharide (AOS)-modified gut microbiota was able to significantly decreased

blood glucose and recover 5 to 10 times the concentration and motility of the subjects' sperm.

Response: Thank the reviewer very much for the very nice comment.

August 9, 2022

Dr. Yong Zhao
CAAS
2 Yuanmingyuan Western Road
Beijing
China

Re: Spectrum01423-22R1 (Gut-testis axis: microbiota prime metabolome to increase sperm quality in young type 2 diabetes)

Dear Dr. Yong Zhao:

Link Not Available

Sincerely,

Henning Seedorf

Journals Department
Reviewer comments:

Reviewer #1 (Comments for the Author):

Thank you for revising the manuscript.

There are a few remarks on changes made and some minor edits. There is also a comment that was not addressed. You will find all suggestions below:

Unaddressed comments

Comment #3: Fig 2a-c - gut microbiota of control mice data are missing Response: Thank the reviewer very much for the very nice comment.

Further comments on points addressed

Comment #1: Test effectivity of T2D treatment with FMT on male fertility by performing fertility tests such as examining how many pregnancies come to term and offspring health status. This is particularly important as the authors state that T2D is linked to various symptoms of male infertility, not confined to sperm count and motility. Response: Thank the reviewer very much for the very nice comment. In current study, we did not detect the pregnant rate or number of living pups/litter. We plan to perform experiments to determine it in the future. Thanks.

Please refer to this in the discussion stating that to claim improve sperm quality one must indeed document a significant increase in pregnancies that come to term.

Comment 4: Across the figures, very few data points appear at times. For example, FigS1, S4. All the analyses were done with more than three samples per group.

Three samples per group is a minimum number that can be stuck to when resources are scarce, and the data points are not widely spread.

Comment #11: Fig 3D stats between control and HS is needed here again. The reason is that one can establish the phenotype of T2D and therefore the effects of any treatment on T2D ONLY when there is significance between control and HS (untreated T2D model). Response: Thank the reviewer very much for the very nice comment. We added stats for Fig. 1c and revised the text as the reviewer suggested. Thanks.

Please remove Fig 3D as the difference between HS and Con is not significant, so any data is not representative of the model the authors present.

Comment #12: Line 191 "Similarly, blood retinoic acid was decreased in HS, however, it was increased in A10-FMT (Fig. 3m)." needs to be rephrased to reflect the lack of significance. One can state trends and then in Fig4 where the protein synthesis of retinoic acid is discussed, remind the reader of Fig 3m. Response: Thank the reviewer very much for the very nice comment. We revised the text as the reviewer suggested. Thanks.

Line 193 Please add the following:

"It should be noted that in some cases the data were not significant between Con and HS (Fig 3k-m), which suggests that the changes observed may not be representative for T2D."

Comment #14: Fig 4J stats need to be provided for the control group. All inferences have to be made on the control regarding the presence of the phenotype. Fig 4i does not show significant increase in GPX1 between the HS and A10-FMT groups, if I am interpreting the "ab" labelling correctly. The particular choice of statistical strength labeling is confusing. The graphs can be broken down to simple graphs with data points, error bars and asterisks for significance. Response: Thank the reviewer very much for the very nice comment. We added stats for Fig. 4j and revised the text as the reviewer suggested. Yes, the data were significant between Con and HS for GPX1. We revised the text. We have tried to make the graph clear for Fig. 4 i, however, if all the data shown in one graph, it is hard to make it very simple. Thank the review for the understanding. Thanks.

Line 223 Please edit the text to:

"Con and HS for some of the compounds (Fig. 5a, c, f, l, k, n, o, p)."

AND

"... however, the data were not significant between Con and HS which suggests that the changes observed may not be representative for T2D."

Comment #15: Fig5 stats need to be provided for the control group throughout this figure or the inferences made are not as telling as stated in the paper. Figures 5d, e, h, l & m can be used to infer effectivity of the treatment. A potential treatment cannot just reflect a difference to the pathology but has to reflect similarity to the WT because there can also be adverse effects and side effects to each treatment which need to be made known for consideration. Response: Thank the reviewer very much for the very nice comment. We added stats for Fig. 5 as the reviewer suggested, and revised the text as the reviewer suggested. We understand what the reviewer concerned. We are trying to make the result clear and correct to the readers as the reviewer suggested. Thanks.

Line 233 please edit "which suggests that the changes observed may not be representative for T2D"

Comment #17: The statement "HS treatment significantly decreased these hormones while A10-FMT led to a recovery (Fig. 6a-

c)." is not true for Fig6c. Response: Thank the reviewer very much for the very nice comment. It was revised as the reviewer suggested. Thanks.

Line 239. Please edit text to:

"HS treatment significantly decreased androsterone while it was in an increasing trend in A10-FMT (Fig. 6c) which however may not be relevant to T2D given that the data were not significant between Con and HS."

Comment #20: Fig 7f No significance between control and HS for TP1 and PIWIL1. Please rephrase in the text (lines 257-8). I would also recommend trying tubulin as a control because it seems that actin was overloaded and may not have served as an adequate loading control. Response: Thank the reviewer very much for the very nice comment. It was revised as the reviewer suggested. We have also tried GAPDH as the loading control. And the data were similar. Thanks.

Actin appears overloaded. Since the GAPDH has also already been used, it can be placed in a supplemental figure.

General comments

- line 206: too many spaces between "Acaa2" and "Acox1".
- Line 351-2: "was freshly prepared every day and was delivered"
- line 372: please spell out IP for the first time in the manuscript "intraperitoneally (IP)" - this can be done at line 130 where it is mentioned that streptozotocin is injected
- Line 226 - "improved liver function"
- Line 227 - "to explore the effects of A10-FMT"
- Line 257 "to recover spermatogenesis, sperm concentration and motility"
- Line 258 "This appears to be an actual effect" - remove

Reviewer #3 (Comments for the Author):

The authors followed all the recommendations made by the reviewers, and both the text and the analyses were greatly improved.

Staff Comments:

Preparing Revision Guidelines

Please return the manuscript within 60 days; if you cannot complete the modification within this time period, please contact me. If you do not wish to modify the manuscript and prefer to submit it to another journal, please notify me of your decision immediately so that the manuscript may be formally withdrawn from consideration by Microbiology Spectrum.

Response to reviewer's comments

First, we would like to thank the editors and reviewers very much for the comments.

Reviewer(s)' Comments to Author:

Reviewer #1 (Comments for the Author): Thank you for revising the manuscript. There are a few remarks on changes made and some minor edits. There is also a comment that was not addressed. You will find all suggestions below:

Unaddressed comments

Comment (R2) #1: Comment #3: Fig 2a-c - gut microbiota of control mice data are missing Response: Thank the reviewer very much for the very nice comment.

Response: Thank the reviewer very much for the very nice comment. First, I am sorry I did not explain it well last time. The data have been added into Fig2a-c as the reviewer suggested. Thanks.

Further comments on points addressed

Comment (R2) #2: Comment #1: Test effectivity of T2D treatment with FMT on male fertility by performing fertility tests such as examining how many pregnancies come to term and offspring health status. This is particularly important as the authors state that T2D is linked to various symptoms of male infertility, not confined to sperm count and motility. Response: Thank the reviewer very much for the very nice comment. In current study, we did not detect the pregnant rate or number of living pups/litter. We plan to perform experiments to determine it in the future. Thanks. Please refer to this in the discussion stating that to claim improve sperm quality one must indeed document a significant increase in pregnancies that come to term.

Response: Thank the reviewer very much for the very nice comment. We added the statement in the discussion section as the reviewer suggested “In current investigation, sperm motility and concentration were increased by A10-FMT compared to T2D (HS group), however, the pregnancy rate was not determined”. Thanks.

Comment (R2) #3: Comment 4: Across the figures, very few data points appear at times. For example, FigS1, S4. All the analyses were done with more than three samples per group.

Three samples per group is a minimum number that can be stuck to when resources are scarce, and the data points are not widely spread.

Response: Thank the reviewer very much for the very nice comment. The reviewer is right. We totally agree. For gut microbiota and metabolites analysis, there were at least 10 samples per group. For RNA-seq analysis, there were three samples per group. As shown in Fig. S4b, all the groups were separated well. For some of the other experiments, there were at least three samples per group. Thanks.

Comment (R2) #4: Comment #11: Fig 3D stats between control and HS is needed here again. The reason is that one can establish the phenotype of T2D and therefore the effects of any treatment on T2D ONLY when there is significance between control and HS (untreated T2D model). Response: Thank the reviewer very much for the very nice comment. We added stats for Fig. 1c and revised the text as the reviewer suggested. Thanks.

Please remove Fig 3D as the difference between HS and Con is not significant, so any data is not representative of the model the authors present.

Response: Thank the reviewer very much for the very nice comment. Fig 3d was deleted as the reviewer suggested. Thanks.

Comment (R2) #5: Comment #12: Line 191 "Similarly, blood retinoic acid was decreased in HS, however, it was increased in A10-FMT (Fig. 3m)." needs to be rephrased to reflect the lack of significance. One can state trends and then in Fig4 where the protein synthesis of retinoic acid is discussed, remind the reader of Fig 3m. Response: Thank the reviewer very much for the very nice comment. We revised the text as the reviewer suggested. Thanks.

Line 193 Please add the following:

"It should be noted that in some cases the data were not significant between Con and HS (Fig 3k-m), which suggests that the changes observed may not be representative for T2D."

Response: Thank the reviewer very much for the very nice comment. It was added as the reviewer suggested. Thanks.

Comment (R2) #6: Comment #14: Fig 4J stats need to be provided for the control group. All inferences have to be made on the control regarding the presence of the phenotype. Fig 4i does not show significant increase in GPX1 between the HS and A10-FMT groups, if I am interpreting the "ab" labelling correctly. The particular choice of statistical strength labeling is confusing. The graphs can be broken down to simple graphs with data points, error bars and asterisks for significance. Response: Thank the reviewer very much for the very nice comment. We added stats for Fig. 4j and revised the text as the reviewer suggested. Yes, the data were significant between Con and HS for GPX1. We revised the text. We have tried to make the graph clear for Fig. 4 i, however, if all the data shown in one graph, it is hard to make it very simple. Thank the review for the understanding. Thanks.

Line 223 Please edit the text to:

"Con and HS for some of the compounds (Fig. 5a, c, f, I, k, n, o, p)."

AND

"... however, the data were not significant between Con and HS which suggests that the changes observed may not be representative for T2D."

Response: Thank the reviewer very much for the very nice comment. It was added as the reviewer suggested. Thanks.

Comment (R2) #7: Comment #15: Fig5 stats need to be provided for the control group throughout this figure or the inferences made are not as telling as stated in the paper. Figures 5d, e, h, l & m can be used to infer effectivity of the treatment. A potential treatment cannot just reflect a difference to the pathology but has to reflect similarity to the WT because there can also be adverse effects and side effects to each treatment which need to be made known for consideration. Response: Thank the

reviewer very much for the very nice comment. We added stats for Fig. 5 as the reviewer suggested, and revised the text as the reviewer suggested. We understand what the reviewer concerned. We are trying to make the result clear and correct to the readers as the reviewer suggested. Thanks.

Line 233 please edit "which suggests that the changes observed may not be representative for T2D"

Response: Thank the reviewer very much for the very nice comment. It was added as the reviewer suggested. Thanks.

Comment (R2) #8: Comment #17: The statement "HS treatment significantly decreased these hormones while A10-FMT led to a recovery (Fig. 6a-c)." is not true for Fig6c. Response: Thank the reviewer very much for the very nice comment. It was revised as the reviewer suggested. Thanks.

Line 239. Please edit text to:

"HS treatment significantly decreased androsterone while it was in an increasing trend in A10-FMT (Fig. 6c) which however may not be relevant to T2D given that the data were not significant between Con and HS."

Response: Thank the reviewer very much for the very nice comment. It was revised as the reviewer suggested. Thanks.

Comment (R2) #9: Comment #20: Fig 7f No significance between control and HS for TP1 and PIWIL1. Please rephrase in the text (lines 257-8). I would also recommend trying tubulin as a control because it is seems that actin was overloaded and may not have served as an adequate loading control. Response: Thank the reviewer very much for the very nice comment. It was revised as the reviewer suggested. We have also tried GAPDH as the loading control. And the data were similar. Thanks.

Actin appears overloaded. Since the GAPDH has also already been used, it can be placed in a supplemental figure.

Response: Thank the reviewer very much for the very nice comment. We did a new blotting. The Fig. 7e was revised as the reviewer suggested. Thanks.

Comment (R2) #10: General comments

- line 206: too many spaces between "Acaa2" and "Acox1".
- Line 351-2: "was freshly prepared every day and was delivered"
- line 372: please spell out IP for the first time in the manuscript "intraperitoneally (IP)" - this can be done at line 130 where it is mentioned that streptozotocin is injected
- Line 226 - "improved liver function"
- Line 227 - "to explore the effects of A10-FMT"
- Line 257 "to recover spermatogenesis, sperm concentration and motility"
- Line 258 "This appears to be an actual effect" – remove

Response: Thank the reviewer very much for the very nice comment. All these points have been revised as the reviewer suggested. Thanks.

September 20, 2022

Dr. Yong Zhao
CAAS
2 Yuanmingyuan Western Road
Beijing
China

Re: Spectrum01423-22R2 (Gut-testis axis: microbiota prime metabolome to increase sperm quality in young type 2 diabetes)

Dear Dr. Yong Zhao:

Your manuscript has been accepted, and I am forwarding it to the ASM Journals Department for publication. You will be notified when your proofs are ready to be viewed.

Sincerely,

Henning Seedorf
Editor, Microbiology Spectrum

Journals Department
Supplemental Dataset: Accept
Supplemental Material: Accept
Supplemental Dataset: Accept